# GWAS of thyroid stimulating hormone highlights pleiotropic effects and inverse association with thyroid cancer

Wei Zhou [ID] et al.[#]

Thyroid stimulating hormone (TSH) is critical for normal development and metabolism. To better understand the genetic contribution to TSH levels, we conduct a GWAS meta-analysis at 22.4 million genetic markers in up to 119,715 individuals and identify 74 genome-wide significant loci for TSH, of which 28 are previously unreported. Functional experiments show that the thyroglobulin protein-altering variants P118L and G67S impact thyroglobulin secretion. Phenome-wide association analysis in the UK Biobank demonstrates the pleiotropic effects of TSH-associated variants and a polygenic score for higher TSH levels is associated with a reduced risk of thyroid cancer in the UK Biobank and three other independent studies. Two-sample Mendelian randomization using TSH index variants as instrumental variables suggests a protective effect of higher TSH levels (indicating lower thyroid function) on risk of thyroid cancer and goiter. Our findings highlight the pleiotropic effects of TSH-associated variants on thyroid function and growth of malignant and benign thyroid tumors.

---

[#]A list of authors and their affiliations appears at the end of the paper.

Normal thyroid function is essential for proper growth and development, and for metabolic functions. Approximately 20 million people in the United States are affected by thyroid disorders and 12% of the population is expected to develop thyroid conditions over their life span[1]. Thyroid-stimulating hormone (TSH) is secreted by the pituitary gland and stimulates the growth of the thyroid gland, and its synthesis and secretion of thyroid hormones. These include thyroxine (T4), most of which is converted to its more bioactive form, 3,3′,5-triiodothyronine (T3). TSH levels are negatively regulated by T3 and T4, and lower or higher levels than the reference range, respectively, usually suggest that the thyroid gland is overactive as in primary hyperthyroidism or underactive as in primary hypothyroidism. The complex inverse relationship between TSH and thyroid hormones means TSH is a more sensitive marker of thyroid status, a feature that has been used to identify individuals with thyroid dysfunction[2].

Thyroid disorders affect multiple organs and are associated with a range of clinical consequences, including an increased risk of metabolic disorders and cardiovascular mortality[3–6]. Over the past few decades, a steady increase of the incidence rates of non-medullary thyroid cancer (henceforth referred as thyroid cancer) has been observed in most areas of the world, including in Europe[7]. Previous studies have led to inconsistent conclusions on the relationship between TSH levels and thyroid cancer risk[8]. Several studies have observed an association between low TSH levels, which can occasionally occur as a consequence of autonomous thyroid nodules and an increased risk of thyroid cancer[8–13]. In contrast, several studies have indicated that TSH promotes the growth of thyroid cancers[8,14–16], which has led to the recommendation to lower TSH levels among people with thyroid cancer to reduce the risk of cancer recurrence. Two initial genome-wide association studies (GWAS) identified five significant loci for thyroid cancer in Europeans and the risk alleles of all five loci have been associated with decreased TSH levels[17,18]. In contrast, a more recent GWAS identified five additional loci associated with thyroid cancer, none of which were even nominally associated with TSH[19]. A recent two-sample Mendelian randomization study suggested a causal inverse association between TSH levels and overall cancer risk, including thyroid cancer[20]. Additional studies are needed to clarify the role of TSH and TSH-associated variants in thyroid cancer.

Twin studies have shown TSH levels are moderately heritable, with estimates up to 65%[21]. Previous TSH GWAS studies have identified 46 independent TSH-associated loci[17,22,23], accounting for 9.4% of TSH variance, thus leaving a large proportion of the TSH heritability unexplained[23,24]. With the goal of identifying the missing genetic components for TSH to further understand its underlying genetic architecture and impact on thyroid cancer, we perform a GWAS meta-analysis for TSH levels on the population-based Nord-Trøndelag Health Study (HUNT study) ($N = 55{,}342$)[25], Michigan Genomics Initiative[26] (MGI, $N = 10{,}085$), and the ThyroidOmics consortium (up to $N = 54{,}288$ samples)[23].

To investigate the genetic relationship between TSH and thyroid cancer, and other human diseases, we examine phenome-wide associations in the UK Biobank (UKBB)[27] for TSH-associated index variants. We also conduct phenome-wide association tests for the polygenic scores (PGS) of TSH in the UKBB and the FinnGen study. We observe an association between high TSH PGS and low thyroid cancer risk, and replicate that observation in two other study populations from Columbus, USA and Iceland[19]. To evaluate the potential causality of TSH on thyroid cancer, we perform a two-sample Mendelian Randomization analysis using the TSH-associated top association signals as instrumental variables and the thyroid cancer GWAS results on

736,049 individuals (4146 cases and 731,903 controls) from a meta-analysis of UKBB[27,28], MGI[26], and results from a previous meta-analysis for thyroid cancer based on a Icelandic data set from deCODE (referred to as deCODE in this manuscript), as well as four other case–control data sets with European ancestry as reported in Gudmundsson et al.[19].

## Results

**Discovery of genetic loci for TSH.** We identified 74 loci associated with TSH (Table 1, Supplementary Data 1 and 2, and Supplementary Fig. 1) in our meta-analysis of the HUNT study ($N = 55{,}342$), the MGI biobank ($N = 10{,}085$), and the ThyroidOmics consortium[23] (up to $N = 54{,}288$). Twenty-eight of the 74 loci have not been previously reported for TSH[17,22,23] (Table 1). To identify secondary independent association signals, we performed stepwise conditional analysis within each locus using GCTA-COJO[29] based on GWAS summary statistics from the meta-analysis of HUNT, MGI, and ThyroidOmics, and the linkage disequilibrium (LD) correlation between variants estimated in HUNT. We observed additional associations in 1 novel TSH locus (B4GALNT3) and 11 previously known TSH loci (Table 1 and Supplementary Data 2). In total, 99 independent top variants have been identified at the 74 loci, explaining 13.3% of the variance of TSH levels.

Despite having only moderate effect sizes, top variants in several novel TSH loci point to nearby genes with a known or suspected link to thyroid function (Table 1 and Supplementary Fig. 2). An intronic variant (rs10186921) in the thyroid adenoma-associated gene THADA was identified to be associated with TSH. THADA has been identified as a somatic mutated/rearranged gene in papillary thyroid cancer[30] and observed to be truncated in thyroid adenoma[31]. Although THADA is known to play a role in cold adaptation, obesity, and type 2 diabetes, its role in thyroid function remains elusive[32,33]. A rare missense variant rs145153320 in gene B4GALNT3 is associated with TSH (minor allele frequency in HUNT ($MAF_{HUNT}$) = 0.25%, effect $size_{HUNT}$ = 0.49 standard deviation (SD), 95% confidence interval (CI) = 0.35–0.63 SD, $P\text{-value}_{HUNT} = 1.00 \times 10^{-11}$) and is 11 times more frequent in the Norwegian HUNT samples than in other non-Finnish Europeans[34]. The WNK1-B4GALNT3 gene fusion has been identified in papillary thyroid carcinoma[35].

Two novel independent rare coding variants with effect sizes larger than one SD were identified in the known TSH locus TSHR, which encodes the TSH receptor. Both variants were only observed in HUNT. The rare missense variant TSHR p.R609Q (rs139352934, $MAF_{HUNT}$ = 0.20%, effect $size_{HUNT}$ = 1.1 SD, 95% CI = 0.94–1.26 SD) is the most significant variant in the locus ($P\text{-value}_{HUNT}$ = $2.66 \times 10^{-41}$) followed by p.A553T (rs121908872, $MAF_{HUNT}$ = 0.07%, effect $size_{HUNT}$ = 1.63 SD, 95% CI = 1.36–1.90 SD, $P\text{-value}_{HUNT} = 2.79 \times 10^{-32}$). TSHR p.R609Q (rs139352934) is 22 times more frequent in HUNT than in other non-Finnish Europeans[34]. TSHR p.R609Q has been reported to aggregate in a family with non-autoimmune isolated hyperthyrotropinemia[36] and TSHR p.A553T has been previously detected in a family with congenital hypothyroidism[37].

As single-variant association tests may lack power for rare variants (MAF ≤ 0.5%) and to search for genes with multiple rare protein-altering variants, we performed exome-wide gene-based SKAT-O[38] tests as implemented in SAIGE-GENE[39] to identify rare coding variants associated with TSH. We grouped missense and stop-gain variants with MAF ≤ 0.5% and imputation quality score ≥ 0.8 within each gene and tested 10,071 genes with at least two variants. This analysis identified two genes, TSHR and B4GALNT3, as significantly associated with TSH ($P\text{-value} < 2.5 \times 10^{-6}$; Supplementary Table 1 and Supplementary Fig. 3). Rare

**Table 1 Lead variants within 28 novel independent loci associated with TSH identified in the meta-analysis of HUNT, MGI, and ThyroidOmics[23].**

| Locus Index | Chromosome:Position (build 37) | rs ID | Ref. | Alt | Category | Nearest Gene(s) | HUNT + MGI + ThyroidOmics[23] meta-analysis | | | | | | |
|---|---|---|---|---|---|---|---|---|---|---|---|---|---|
| | | | | | | | Freq[a] | Effect[b] | SE | P | N | Direction[c] | Heterogeneity P-value |
| 1 | 1:22513011 | rs12743883 | A | G | Intergenic | WNT4;MIR4418 | 0.592 | 0.031 | 0.005 | 1.00E−10 | 116,445 | +++ | 0.57 |
| 2 | 1:51451499 | rs11583886 | G | A | intergenic | CDKN2C;MIR4421 | 0.306 | −0.035 | 0.005 | 3.83E−12 | 116,445 | −−− | 0.97 |
| 3 | 1:68166425 | rs12029562 | G | A | Downstream | GNG12 | 0.525 | −0.030 | 0.005 | 1.66E−10 | 116,445 | −−− | 0.81 |
| 4 | 1:218685055 | rs2993047 | G | A | ncRNA_intronic | C1orf143;MIR548F3 | 0.592 | −0.033 | 0.005 | 3.71E−12 | 116,445 | −−− | 0.23 |
| 5 | 2:25994220 | rs6721104 | A | C | Intronic | ASXL2 | 0.036 | 0.087 | 0.012 | 4.45E−12 | 117,474 | +++ | 0.98 |
| 6 | 2:43644556 | rs10186921 | C | T | Intronic | THADA | 0.542 | 0.040 | 0.005 | 3.13E−18 | 117,474 | +++ | 0.22 |
| 7 | 2:169554118 | rs62174422 | T | G | Intronic | CERS6 | 0.039 | −0.083 | 0.012 | 2.09E−11 | 117,474 | −−− | 0.56 |
| 8 | 2:242516105 | rs6717283 | A | G | Intergenic | BOK;THAP4 | 0.142 | 0.046 | 0.007 | 1.04E−11 | 116,443 | +++ | 0.82 |
| 9 | 3:188072513 | rs9865818 | A | G | Intronic | LPP | 0.431 | −0.026 | 0.005 | 3.42E−08 | 119,715 | −−− | 0.45 |
| 10 | 4:177705862 | rs4571283 | G | A | Intronic | VEGFC | 0.875 | 0.045 | 0.008 | 2.72E−09 | 107,389 | +?+ | 0.57 |
| 11 | 5:58373418 | rs77994712 | C | G | Intronic | PDE4D | 0.052 | −0.066 | 0.011 | 1.44E−09 | 118,501 | −−− | 0.53 |
| 12 | 6:168819800 | rs751171 | T | C | Intergenic | DACT2;SMOC2 | 0.338 | 0.033 | 0.005 | 1.04E−11 | 119,715 | +++ | 0.66 |
| 13 | 7:2329497 | rs4719486 | G | A | Intronic | SNX8 | 0.419 | −0.027 | 0.005 | 8.44E−09 | 119,715 | −−− | 0.12 |
| 14 | 7:46753491 | rs700750 | C | A | Intergenic | LOC730338;TNS3 | 0.643 | 0.034 | 0.005 | 7.90E−13 | 119,715 | +++ | 0.84 |
| 15 | 8:8323088 | rs2979181 | A | T | Intergenic | PRAG1;CLDN23 | 0.496 | −0.036 | 0.005 | 3.82E−11 | 119,715 | −−− | 0.26 |
| 16 | 8:120112818 | rs72682433 | T | C | Intronic | COLEC10 | 0.097 | 0.042 | 0.008 | 4.22E−08 | 119,715 | +++ | 0.68 |
| 17 | 9:127032607 | rs1045774 | A | G | Intronic | NEK6 | 0.384 | 0.032 | 0.005 | 3.87E−12 | 119,715 | +++ | 0.37 |
| 18 | 12:549670 | rs546738875 | C | G | Intronic | CCDC77 | 0.003 | 0.390 | 0.061 | 1.95E−10 | 65,427 | +−? | 0.11 |
| 18 | 12:570840 | rs7966590 | G | A | Intronic | B4GALNT3 | 0.550 | 0.039 | 0.005 | 5.93E−14 | 117,977 | +++ | 0.12 |
| 18 | 12:665822 | rs145153320 | C | T | Nonsynonymous | B4GALNT3 | 0.003 | 0.489 | 0.073 | 2.04E−11 | 55,342 | +?? | 1.00 |
| 19 | 12:69831694 | rs10878986 | T | C | Intergenic | YEATS4;FRS2 | 0.385 | −0.027 | 0.005 | 2.40E−08 | 119,715 | −−− | 0.77 |
| 20 | 12:111884608 | rs3184504 | T | C | Nonsynonymous | SH2B3 | 0.518 | −0.030 | 0.005 | 7.37E−11 | 119,715 | −−− | 0.88 |
| 21 | 13:111191813 | rs4393429 | T | C | Intronic | RAB20 | 0.256 | −0.030 | 0.005 | 8.75E−09 | 119,715 | −−− | 0.08 |
| 22 | 15:36001394 | rs74888443 | C | T | ncRNA_intronic | DPH6-AS1 | 0.054 | 0.060 | 0.011 | 2.70E−08 | 119,715 | +++ | 0.41 |
| 23 | 17:47418178 | rs35587648 | G | A | Intronic | ZNF652 | 0.383 | 0.032 | 0.005 | 4.53E−12 | 119,715 | +++ | 0.29 |
| 24 | 17:64208285 | rs1801690 | C | G | Nonsynonymous | APOH | 0.059 | −0.065 | 0.010 | 2.76E−11 | 119,715 | −−− | 0.38 |
| 25 | 19:2634823 | rs72978712 | T | C | Intronic | GNG7 | 0.204 | 0.044 | 0.007 | 9.42E−11 | 115,129 | +++ | 0.97 |
| 26 | 19:5572412 | rs10421676 | A | G | Intergenic | PPP6R1;HSPBP1 | 0.598 | 0.027 | 0.005 | 1.05E−08 | 117,475 | +++ | 0.97 |
| 27 | 20:6685377 | rs6085658 | C | T | Intergenic | CASC20;LINC01713 | 0.400 | −0.029 | 0.005 | 1.32E−09 | 119,715 | −−− | 0.64 |
| 28 | 22:31790014 | rs5997969 | T | C | Intergenic | LINC01521;DRG1 | 0.665 | −0.027 | 0.005 | 4.37E−08 | 119,715 | −−− | 0.57 |

aFrequencies are reported with respect to the alternate allele in the combined meta-analysis data set.
bEffect sizes are reported with respect to the alternate allele in the unit of SD of TSH levels.
cEffect direction on TSH of the alternate allele in individual data sets: HUNT, MGI, and ThyroidOmics, respectively. Noted as ? if the variant is missing in the corresponding data set.

variants in both genes associated with TSH were also identified from single-variant analysis. After conditioning on the two rare variants in *TSHR* that were genome-wide significant in the single-variant analysis ($P$-value $< 5 \times 10^{-8}$), rs121908872 and rs139352934, the gene *TSHR* was still exome-wide significant with $P$-value $2.87 \times 10^{-8}$, while *B4GALNT3* was no longer significantly associated with TSH with $P$-value 0.3 after conditioning on the top variant rs145153320.

**Fine-mapping for potentially causal variants among TSH loci**. To identify potentially causal variants at TSH loci, we conducted fine-mapping using SuSiE[40], which estimates the number of causal variants and obtains credible sets of variants with 95% cumulative posterior probability through Iterative Bayesian Stepwise Selection[41] (Supplementary Data 3). The LD matrix used in SuSiE was calculated based on HUNT. We identified eight independent causal variants at the *TSHR* locus by fine-mapping using SuSiE[40] and seven independent association signals by the stepwise conditional analysis (Supplementary Data 2), suggesting allelic heterogeneity at the *TSHR* locus.

In addition, fine-mapping by SuSiE[40] and stepwise conditional analysis identified two association signals in the locus of the thyroglobulin gene (*TG*). *TG* encodes a highly specialized homo-dimeric multidomain glycoprotein for thyroid hormone biosynthesis[27], it is the most highly expressed gene in the thyroid gland and its protein product represents roughly half the protein of the entire thyroid gland[42,43]. The *TG* locus has been reported in a recent TSH GWAS[23]. The 95% credible set for each causal association contains one missense variant that is in strong LD with the most strongly associated intronic variant (Supplementary Table 2 and Supplementary Fig. 4). In the HUNT study, the missense variant *TG* p.G67S (rs116340633, MAF = 1.8%, effect size = −0.26 SD, 95% CI = −0.31–−0.20 SD, $P$-value $= 1.07 \times 10^{-21}$) is in strong LD ($r^2$ = 0.99) with the most strongly associated variant rs117074997 (intronic). At the other association signal, missense variant *TG* p. P118L (rs114322847, MAF = 2.4%, effect size = −0.17 SD, 95% CI = −0.20–−0.14 SD, $P$-value $= 1.87 \times 10^{-26}$) is in strong LD ($r^2$ = 0.92) with the most strongly associated variant rs118039499 (intronic) (Supplementary Table 2 and Supplementary Fig. 4). *TG* p.P118L has been previously detected among familial cases with congenital hypothyroidism[44]. *TG* p.P118L (rs114233847) is significantly associated with nontoxic nodular goiter (odds ratio (OR) = 2.69, 95% CI = 2.05–3.54, $P$-value $= 5.8 \times 10^{-12}$) in the UKBB[27,28], while the association of TG p.G67S (rs116340633) with nontoxic nodular goiter is less significant (OR = 1.63, 95% CI = 1.08-2.49, $P$-value $= 2.1 \times 10^{-2}$). *TG* p.P118L has been previously detected in patients with sporadic congenital hypothyroidism in a Finnish cohort[44].

**Functional follow-up of missense variants in the gene TG**. We performed site-directed mutagenesis studies to investigate the impact on the protein expression of TG of the two independent missense variants, both located in the highly conserved Tg1 domain of unclear function. The protein encoded by the human *TG* is conserved in mice, with nearly perfect conservation of all critical amino acid residues, including those that maintain the protein structure and hormone synthesis[45]. A cDNA encoding wild-type mouse Tg (mTg-WT) expressed in 293T cells has normal synthesis and secretion of thyroid hormones[46]. We then introduced the observed human *TG* variants (rs116340633 and rs114322847) into the mTg cDNA. 293T cells were either untransfected or transfected with pcDNA3.1 in which a cyto-megalovirus promoter drives expression of mTg-WT or the p. P118L or p.G67S Tg variants (mature Tg numbering). Then, we examined the intracellular vs. secreted levels of the mTg-WT and

these two human Tg variants (Tg-p.P118L and Tg-p.G67S). Transfected cells were incubated overnight and the culture medium and cell lysates were analyzed by SDS-polyacrylamide gel electrophoresis (PAGE) and immunoblotting with anti-Tg anti-body. The experiment was independently repeated three times and the results analyzed in a manner that is independent of transfection efficiency. On average, 74.6% of the total expressed WT form of mTg was recovered in the media and extracellular : intracellular (M/C) ratio of mTg was 2.94:1, as expected (between 3:1 and 4:1) (Fig. 1). Compared with the WT, the Tg-P118L variant showed a significant reduction in the M/C ratio 0.6:1 ($P$-value = 0.0051) and the Tg-G67S variant also showed a significant reduction in the M/C ratio 1.96:1 ($P$-value = 0.0095).

**Prioritization of TSH genes, pathways, and tissues**. To further understand the biology underlying TSH associations, we prioritized associated genes, tissues, and cell types in which TSH genes are likely to be highly expressed using Data-driven Expression-Prioritized Integration for Complex Traits (DEPICT)[47] based on 161 loci with TSH association $P$-value cutoff $1 \times 10^{-5}$ and clumped based on LD in HUNT. As expected, the membranes and thyroid gland are the most strongly associated tissues followed by tissues from the digestive system (ileum, gastrointestinal tract, pancreas, and colon), respiratory system (lung) and accessory organs for eyes (conjunctiva, eyelids, and anterior eye), although none of the tissues reached the Bonferroni significant threshold ($P$-value $< 0.05/209$) or have false discovery rate (FDR) $< 0.05$ (Supplementary Data 4). Based on functional similarity to other genes among TSH loci, 70 genes at the TSH-associated loci were prioritized by DEPICT with FDR $\leq 0.01$ (Supplementary Data 5), among which the prioritized genes *ZFP36L2*, *B4GALNT3*, *PPP1R3B*, *FAM109A*, *GNG12*, *GADD45A*, *BMP2*, *VEGFC*, *LPP*, and *MAL2* were at the novel TSH loci identified in our meta-analysis (Table 1). In addition, among 14,461 reconstituted gene sets, 56 gene sets were enriched among TSH loci with FDR $< 0.01$. The most significantly enriched one is the CTSD PPI subnetwork, followed by gene sets for regulation of phosphorylation (Supplementary Data 6).

**Pleiotropic effects of TSH loci**. To explore the pleiotropic effects of the TSH loci, we examined associations of the 95 non-human leukocyte antigen (HLA) independent TSH top variants with 1283 human diseases (PheCodes)[27,28,48] and 274 continuous traits (http://www.nealelab.is/uk-biobank) in the UKBB (variants 23:3612081 and rs121908872 are not available in the UKBB). Due to the strong associations between HLA variants and auto-immune diseases[49], we excluded two HLA variants associated with TSH (rs1265091 and rs3104389) in the analysis for pleio-tropic effects. We identified significant ($P$-value $< 5 \times 10^{-8}$) pleiotropic association for 18 out of 95 non-HLA variants across 31 disease phenotypes (Supplementary Fig. 5a and Supplementary Data 7), including thyroid disorders, diabetes, cardiovascular disease, digestive system disorders, asthma, and cataracts. In addition, 34 non-HLA variants were significantly associated ($P$-value $< 5 \times 10^{-8}$) with one or more quantitative traits, including body mass index, lung function measurements, metrics of bone density, spherical power/meridian measurements, and blood cell counts (Supplementary Fig. 5b and Supplementary Data 8). TSH-increasing alleles at one or more loci were associated with an increased risk of cardiovascular disease, smaller body size, reduced bone mineral density, decreased lung function, and an increased risk of hypothyroidism and a decreased risk of goiter. These results are generally consistent with previous studies[23]. We also examined the associations between the TSH index variants and free thyroxine levels in the ThyroidOmics

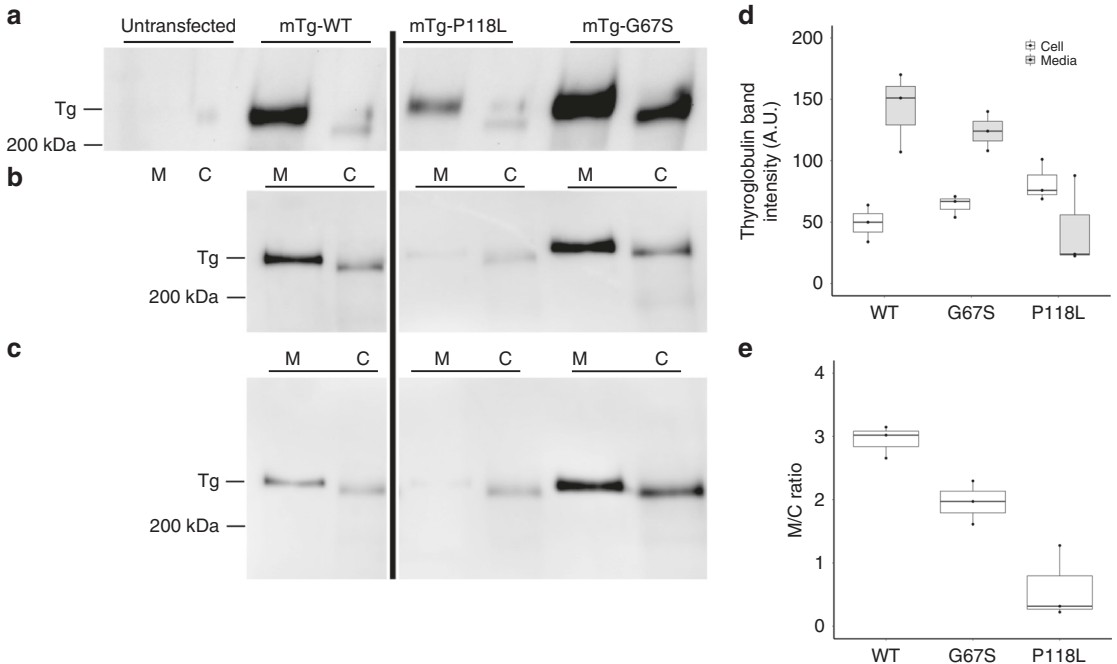

**Fig. 1 Both the *TG*-P118L and *TG*-G67S point mutants exhibit a secretion defect. a–c** Three independent replicate experiments. Western blotting of TG in 293T cells that were either untransfected (**a**, no detectable bands) or transfected with constructs encoding mouse TG wild type (WT) or P118L or G67S point mutants (in the pcDNA3.1 background, in which the CMV promoter drives the respective cDNA expression). Serum-free media (M) were collected overnight and the cells (C) were lysed. Equal volumes of media and cells were analyzed by SDS-PAGE, electrotransfer to nitrocellulose, and immunoblotting with anti-Tg-specific antibodies. Full scans of western blotting are presented in Supplementary Fig. 14. From scanning densitometry, **d** shows the content of thyroglobulin (and its variants) intracellularly and in the secretion. **e** The extracellular : intracellular (M/C) ratio of each construct. **d**, **e** Three independent replicate experiments. All boxplots in **d** and **e** indicate median (center line), 25th and 75th percentiles (bounds of box), and minimum and maximum (whiskers).

consortium[23]. Out of 91 TSH-associated variants, for which association results with free thyroxine were available, 70 (77%) have TSH-lowering alleles associated with higher free thyroxine levels (Supplementary Data 9) ($P$-value$_{binomial}$ = $2.51 \times 10^{-7}$).

We further examined the association with thyroid cancer for TSH index variants. We meta-analyzed UKBB[27,28] and a previous meta-analysis of deCODE and four other case–control data sets[19] for thyroid cancer in 3359 thyroid cancer cases and 694,949 controls and examined 94 out of 99 TSH non-HLA index variants that are available in the meta-analysis for thyroid cancer (Supplementary Data 9). The TSH-increasing alleles of 63 out of 94 TSH-associated variants (67%) were associated with reduced thyroid cancer risk (Supplementary Data 9 and Supplementary Fig. 6a and 6b) ($P$-value$_{binomial}$ = $1.26 \times 10^{-3}$). Eighteen out of the 94 TSH-associated variants tested (19%) were at least nominally associated with thyroid cancer ($P < 0.05$) ($P$-value$_{binomial}$ = $1.18 \times 10^{-9}$). For 16 out of the 18 TSH-associated variants, the TSH-increasing alleles were associated with reduced thyroid cancer risk ($P$-value$_{binomial}$ = $1.31 \times 10^{-3}$, Supplementary Data 9 and Supplementary Fig. 6c, d). Moreover, when we examined alleles that predisposed to thyroid cancer[17–19], 9 out of 11 had a consistent direction of effect towards lower TSH ($P$-value$_{binomial}$ = 0.065). Of the six thyroid cancer risk alleles that were at least nominally associated with TSH level ($P < 0.05$), all six variants were associated with lower TSH ($P$-value$_{binomial}$ = 0.03) (Supplementary Data 10 and Supplementary Fig. 7).

**Associations of polygenic scores of TSH with other phenotypes.** Although individual TSH variants may exhibit pleiotropic effects, it is also possible that the cumulative effects of TSH-modifying genetic variants may lead to disease. Therefore, we constructed

PGS from the 95 independent non-HLA TSH top variants (rs1265091 and rs3104389 are HLA variants and rs121908872 and 23:3612081 were not in UKBB) and examined their association with the 1283 human diseases constructed from International Classification of Diseases (ICD) codes in the UKBB[27,28,48]. As in the pleiotropy analysis, we excluded the two HLA variants in the PGS calculation to study the cumulative genetic effects of TSH-associated variants in non-HLA regions with human diseases. The TSH PGS was significantly associated with 10 phenotypes ($P$-value < $3.9 \times 10^{-5}$, Bonferroni correction for 1283 phenotypes), including an increased hypothyroidism risk and decreased risk of goiter, thyrotoxicosis and hyperhidrosis (Supplementary Data 11 and Supplementary Fig. 8). We also evaluated the phenotypic variance (Nagelkerke's $r^2$)[50] explained by TSH PGS for 596 phenotypes in the UKBB that have at least 500 cases in 280,943 unrelated white British samples (Supplementary Data 12). The phenotypes with highest $r^2$ were nontoxic nodular goiter ($r^2$ = 0.96%), secondary hypothyroidism ($r^2$ = 0.46%), and thyrotoxicosis with or without goiter ($r^2$ = 0.16%). In FinnGen, we also observed that high TSH PGS was associated with high risk of hypothyroidism and low risk of goiter. High TSH PGS in FinnGen was marginally associated with an increase in risk of depression (OR = 1.03 (per SD of TSH PGS), 95% CI = 1.01–1.05, $P$-value = $1.7 \times 10^{-3}$) and a reduced risk of pregnancy hypertension (OR = 0.95 (per SD of TSH PGS), 95% CI = 0.92–0.98, $P$-value = $9 \times 10^{-4}$). The phenome-wide association results for the TSH PGS in FinnGen are shown in Supplementary Data 13 and Supplementary Fig. 9. Depressive symptoms and hypertension during pregnancy have been observed to be clinically associated with hypothyroidism and thyroid dysfunction[51–53], respectively. However, their genetic associations have not been extensively studied.

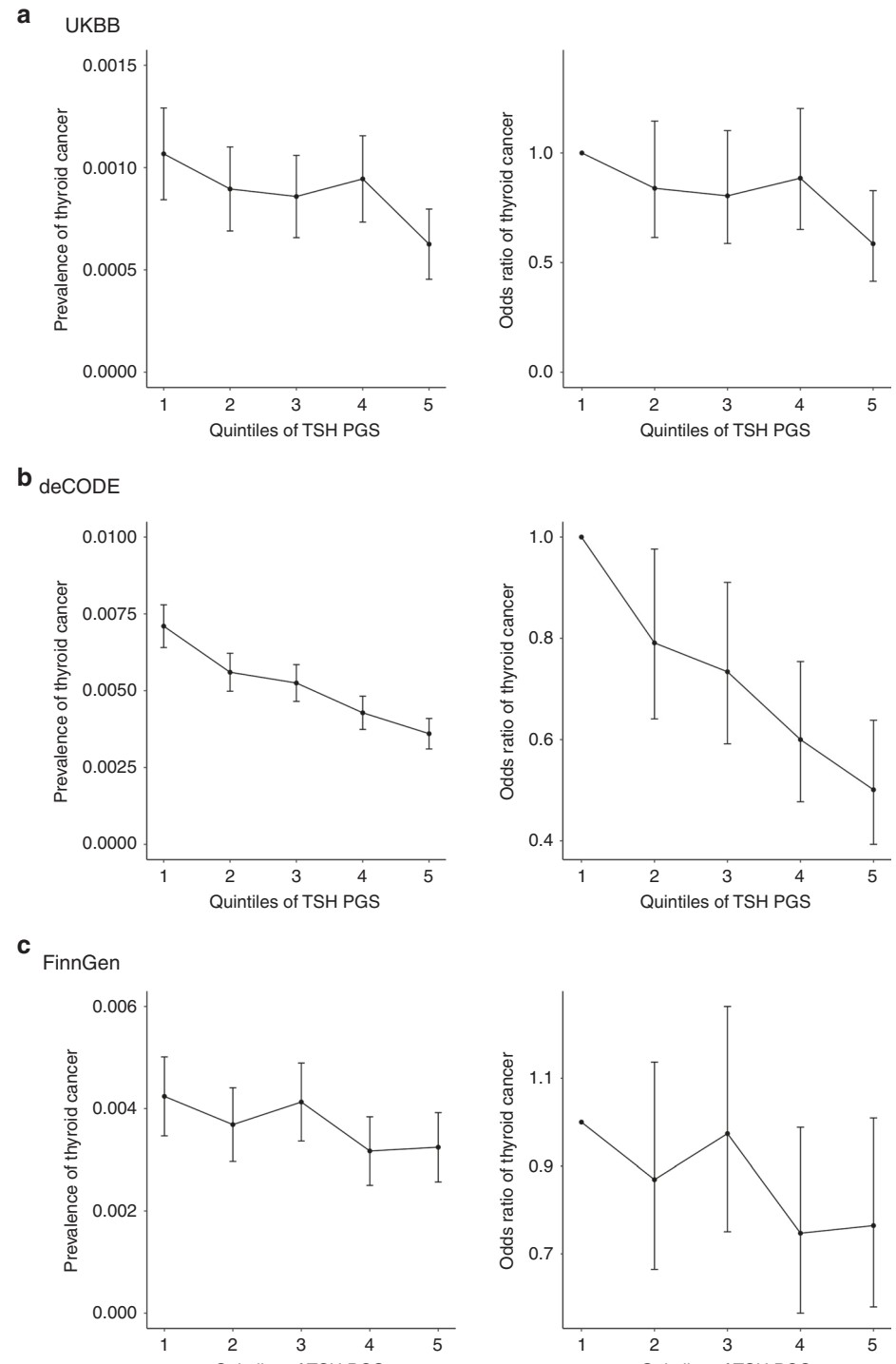

**Fig. 2 The risk of thyroid cancer is lower for individuals with genetically predicted higher TSH levels.** Plots of thyroid cancer prevalence by quintiles of TSH PGS (left) and odds ratio of thyroid cancer in relation to the lowest quintile (right) in data sets UKBB (**a**; N case = 358, N control = 407,399), deCODE (**b**; N case = 1003, N control = 278,991)[19], and FinnGen (**c**; N case = 501, N control = 135,137). N: sample size. N case: sample size of cases. N control: sample size of controls. Error bars represent 95% confidence intervals.

In the UKBB, TSH PGS was significantly associated with a decreased risk of thyroid cancer (OR = 0.86 (per SD of TSH PGS), 95% CI = 0.78–0.96, $P$-value = $5.8 \times 10^{-3}$, Supplementary Data 11 and Fig. 2). Compared with the rest of the samples, the OR (95% CI) for thyroid cancer of individuals with TSH PGS in the lowest quintile was 1.29 (1.01–1.64) and the OR for thyroid cancer of individuals with TSH PGS in the highest quintile was 0.66 (0.49–0.89), suggesting

the protective effects of TSH-increasing genetic variants on thyroid cancer risk.

We successfully replicated the association between high TSH PGS and low thyroid cancer risk in study populations from Columbus, USA[19] (OR = 0.19, 95% CI = 0.10–0.37, $P$-value = $2.1 \times 10^{-7}$) and deCODE[19] (OR = 0.11, 95% CI = 0.06–0.21 $P$-value = $1.1 \times 10^{-11}$). We also observed the association in FinnGen (OR = 0.89, 95% CI = 0.82–0.97, $P$-value = $1.0 \times 10^{-2}$),

although the evidence was much less strong. The Columbus study[19] is a case–control study of thyroid cancer (1580 cases and 1628 controls) with much higher thyroid cancer prevalence than the three population-based biobanks: UKBB[30,31] (358 cases and 407,399 controls), FinnGen (501 cases and 135,137 controls), and deCode[19] (1003 cases and 278,991 controls). In Fig. 2, the prevalence of thyroid cancer (left panel) and OR of thyroid cancer (right panel) are plotted against the TSH PGS for the three population-based cohorts (results for Columbus are provided in Supplementary Fig. 10). Similar plots of hypothyroidism and goiter are plotted for UKBB and FinnGen in Supplementary Figs. 11 and 12.

**Mendelian randomization for TSH, thyroid cancer, and goiter.** We investigated a possible causal effect of TSH on thyroid cancer using two-sample Mendelian randomization. Ninety-four non-HLA genetic variants for TSH identified by our meta-analysis of HUNT, MGI and ThyroidOmics were used as instrumental variables (F-statistic for all single nucleotide polymorphisms (SNPs) > 23.72, rs1265091 and rs3104389 are HLA variants, and the summary statistics for thyroid cancer were not available for rs121908872, rs4571283, and 23:3612081). To avoid sample overlap for the TSH and thyroid cancer GWASs, we used effects on TSH estimated by meta-analyzing HUNT and ThyroidOmics to construct the instrumental variable for TSH levels and we meta-analyzed MGI, deCODE, and UKBB for thyroid cancer. We found that a one SD increase in TSH (SD = 1.036 mU/L) was associated with a 45% decreased risk of thyroid cancer (inverse-variance weighted OR 0.55, 95% CI 0.40–0.74, MR-Egger intercept P-value = 0.54). Sensitivity analyses using the penalized weighted median method, the weighted median method and the weighted mode method including all variants are presented in Fig. 3a and Supplementary Data 14. Similar results were observed between methods, with the exception of the weighted mode, which was strongly attenuated. To reduce the possibility that the results were influenced by occult thyroid dysfunction (typically occurring in older age), we repeated the analysis using SNP-TSH effect estimates obtained among those younger than 50 years of age at the time of TSH measurement (Supplementary Data 15). Similar results were observed, except for the weighted mode which was again attenuated towards the null (OR 1.02, 95% CI 0.82–1.27, Supplementary Data 16). Furthermore, there was strong evidence of heterogeneity, suggesting some instruments were invalid. Nevertheless, when repeating the main analysis using MR-PRESSO, which excluded nine variants due to the detection of specific horizontal pleiotropic outlier variants[54], the causal association was similar (MR-PRESSO outlier corrected OR 0.65, 95% CIs 0.55–0.77) (Fig. 3a and Supplementary Data 14). Finally, using only the protein-coding nonsynonymous variant p.P118L in the TG gene (F-statistic = 114.89), we observed a protective effect of increased TSH on thyroid cancer (Wald ratio, OR 0.23, 95% CI 0.09–0.64). To investigate if TSH may also influence the risk of benign thyroid growth disorders, we similarly performed a two-sample Mendelian Randomization analysis between TSH and goiter. The effects on TSH were estimated by a meta-analysis of HUNT and ThyroidOmics (Supplementary Data 1 and 2) and the GWAS results for goiter from the UKBB were used[27,28]. A 1 SD increase in TSH (SD = 1.036 mU/L) was associated with a 72% decreased risk of goiter (inverse-variance weighted OR 0.28, 95% CI 0.20–0.41, MR-Egger intercept P-value = 0.73) (Fig. 3b and Supplementary Data 17).

## Discussion

Meta-analysis of the HUNT study, the MGI biobank and the ThyroidOmics consortium for TSH on up to 119,715 individuals identified 74 TSH loci, of which 28 are previously unreported. All TSH loci reported by previous GWAS studies[17,22,23] are replicated in our meta-analysis. Several novel loci pointed to nearby genes with a known or suspected link to thyroid function. Additional independent signals were identified among several loci based on GWAS results in the meta-analysis and LD information in the HUNT study, including two rare variants rs546738875 and rs145153320 at the *B4GALNT3* locus and two rare missense variants *TSHR* p.A553T (rs121908872) and *TSHR* p.R609Q (rs139352934), which have been observed to be associated with congenital hypothyroidism in previous family studies[36,37]. *TSHR* p.R609Q (rs139352934) is the most strongly associated with TSH in the TSH receptor gene *TSHR* with an effect size greater than one standard deviation of TSH (1.036 mU/L). As these rare variants were only imputed in HUNT, not in MGI or ThyroidOmics, further follow-up to verify the associations is needed. As individual GWAS was conducted on inverse-normal transformed TSH levels before meta-analysis, it is challenging to convert the effect sizes reported by our meta-analysis to actual scales of TSH levels.

Fine-mapping for potential causal variants among TSH loci detected two independent missense variants in the TG gene *TG*: p.G67S and p.P118L. The two variants have a similar frequency (~2%), but p.P118L shows stronger evidence for an association with goiter and with thyroid cancer. Functional experiments demonstrated each of these defects in the TG gene, *p.P118L* and *p.G67S* respectively, causes defective secretion of TG. Further studies are needed to investigate how the protein-altering variants impact TSH levels.

As expected, membranes and thyroid gland were identified as the tissue in which genes from TSH-associated loci are most likely to be highly expressed. Genes *ZFP36L2, B4GALNT3, PPP1R3B, FAM109A, GNG12, GADD45A, BMP2, VEGFC, LPP,* and *MAL2* were prioritized as functional candidates among the novel TSH-associated loci based on functional similarity to other genes at all TSH loci using DEPICT[47].

A PheWAS of the TSH-associated variants in the UKBB demonstrated that TSH-increasing alleles are associated with an increased risk of cardiovascular disease, smaller body size, reduced bone mineral density, decreased lung function and an increased risk of hypothyroidism, but were favorably associated with a decreased risk of goiter. Our results suggest that these variants have pleiotropic effects, although they tend to affect TSH through actions in the thyroid gland.

Phenome-wide association tests in the UKBB and FinnGen for the TSH PGS showed that genetically predicted increased TSH is associated with a high risk of hypothyroidism and a low risk of goiter, thyrotoxicosis, hyperhidrosis, and thyroid cancer. Two-sample Mendelian randomization analyses suggested that lower TSH causes an increased risk of thyroid cancer and goiter. This is an unexpected direction, given that TSH promotes the growth of thyroid cancers[8,14–16]. Nonetheless, it has previously been speculated that lower TSH levels may lead to less differentiation of the thyroid epithelium, which could predispose to malignant transformation[17]. Alternatively, our genetic instrument for TSH may represent a thyroid gland phenotype that influences both TSH (through the negative feedback of thyroid hormones on the pituitary gland) and thyroid growth (increasing the risk of thyroid cancer and goiter) (Supplementary Fig. 13). In that scenario, the effect on thyroid cancer would not be downstream of TSH, and altering TSH levels (e.g., by medication) would not be expected to alter thyroid cancer risk. Tissue enrichment analyses of genes at TSH-associated loci and the observation that TSH-lowering alleles were generally associated with higher free thyroxine levels (Supplementary Data 9) suggest that most TSH-associated variants act primarily on the thyroid gland, where effects on both thyroid function and growth have previously been described for

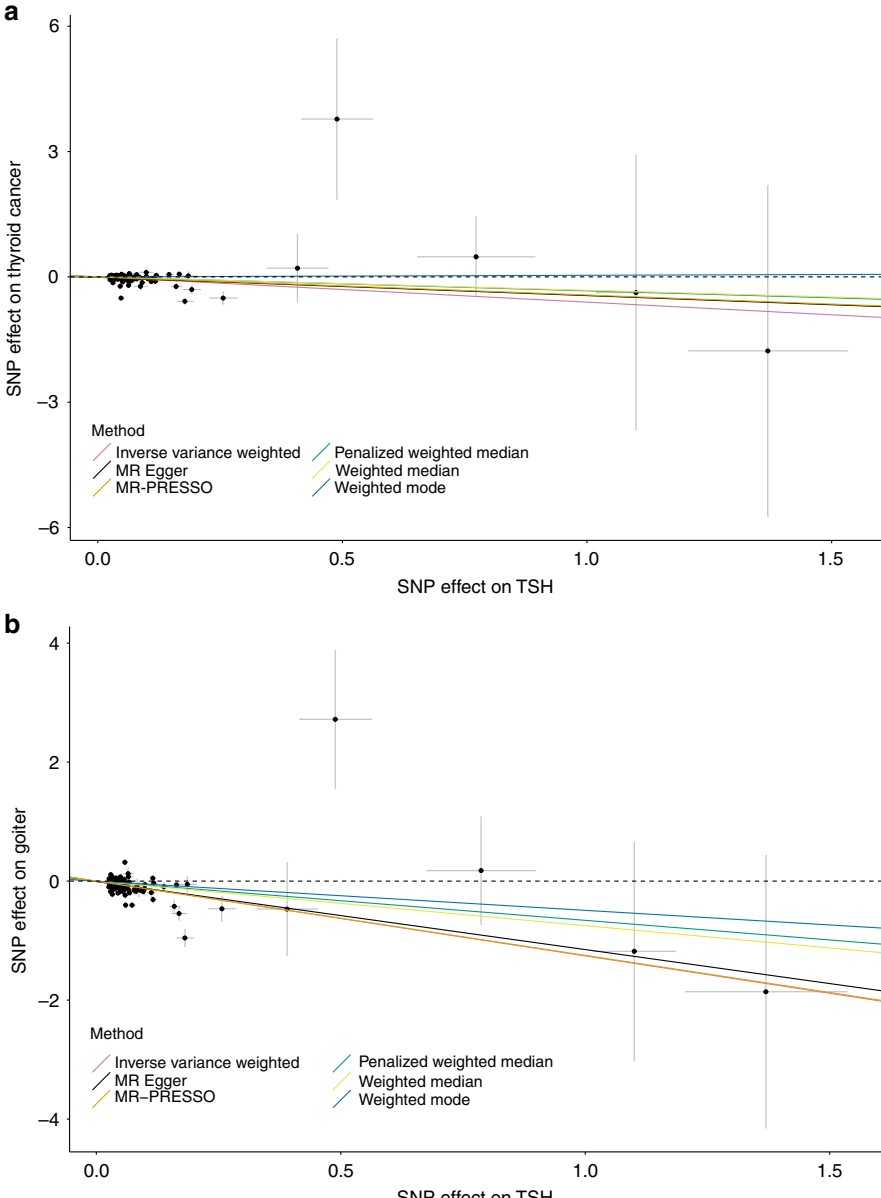

**Fig. 3 Two-sample Mendelian randomization analysis for casual associations between TSH and thyroid cancer and between TSH and goiter. a**. Two-sample MR between TSH and thyroid cancer based on summary statistics from the meta-analysis of HUNT and ThyroidOmics ($n = 106,360$) for TSH and from the meta-analysis of UKBB[27,28], MGI[26], deCODE and four other case–control data sets with European ancestry as reported in Gudmundsson et al.[19] for thyroid cancer association (cases/controls = 4,146/731,903). **b** TSH and goiter based on summary statistics from the same TSH meta-analysis as above and from the GWAS of UKBB[27,28] for goiter association ($N$ case = 1,143, $N$ control = 391,429) The crosshairs on the plots represent the 95% confidence intervals for each SNP-TSH or SNP-outcome association. The variant on the top left corner is the rare nonsynonymous variant *B4GALNT3* p. R724W (rs145153320). *N*: sample size. *N* case: sample size of cases. *N* control: sample size of controls.

*TSHR* mutations[55]. Nonetheless, the Mendelian randomization findings were robust to most of the sensitivity analyses performed to detect and correct for pleiotropy. Restricting the genetic instrument to TSH-lowering variants associated with lower free thyroxine levels could have helped resolve the causal question, but those variants were too few to yield meaningful estimates with current data.

Although our results suggest that interventions elevating TSH levels could potentially reduce thyroid cancer risk, this suggestion lacks evidence from intervention studies. Further, any intervention that increases TSH levels by inhibiting thyroid hormone production would expectedly lead to symptoms and signs of hypothyroidism. Note that phenotype heterogeneity between the

thyroid cancer data sets could have an impact on our results, although associations between higher TSH PGS and reduced thyroid cancer risk were observed in all four study populations. Different approaches were used to curate phenotypes for thyroid cancer, with cases being identified based on ICD codes mapped to PheCodes in the UKBB[28,56], ICD codes combined with population registries in FinnGen and the Icelandic Cancer Registry (ICR) in the Icelandic data set[19], while the papillary or follicular thyroid carcinoma (PTC) cases were histologically confirmed in the Columbia, USA, data set[19].

Our findings suggest that the majority of identified TSH-associated variants act through pathways regulating both thyroid function and thyroid growth. Additional experiments are needed

to fully disentangle the role of TSH-associated variants in thyroid cancer and goiter, such as examining their associations with other thyroid traits (e.g., free thyroxine levels) through large GWASs.

## Methods

**HUNT**. The HUNT study is a longitudinal, repeatedly surveyed, population-based health study conducted in the county of Nord-Trøndelag, Norway, since 1984[16]. Approximately 120,000 individuals have participated through 3 phases of HUNT surveys with phenotype measurements: HUNT1 (1984–86), HUNT2 (1995–97), and HUNT3 (2006–08). About 70,000 HUNT participants with DNA collected have been genotyped using Illumina HumanCoreExome v1.0 and 1.1, and imputed using Minimac3 with a merged reference panel of Haplotype Reference Consortium (HRC)[57] and whole genome sequencing data for 2201 HUNT samples. Variants with imputation $r^2 < 0.3$ were excluded from further analysis. The population is considered to be iodine sufficient[58]. TSH was measured using DELFIA hTSH Ultra from Wallac Oy (Turku, Finland) in HUNT2 and a chemiluminescent microparticle immunoassay on an Architect ci8200 from Abbott (Abbott Ireland, Longford, Ireland; and Abbott Laboratories, Abbott Park, IL) in HUNT3; the two measurement methods yielded similar results[59]. TSH measurements were available for 55,342 individuals who were genotyped (54.4% are females), after excluding those with thyroid disorders based on self-report, blood tests indicating clearly overt hypothyroidism (TSH ≥ 10 mU/L with free T4 < 9 pmol/L; undiagnosed overt autoimmune hypothyroidism was relatively common at the time of HUNT2[59]) and cancer registry data (7.34%). If TSH was measured in both HUNT2 and HUNT3, the earlier HUNT2 measure was used. The mean age at TSH measurement is 51.79 years. The mean TSH in HUNT is 1.659 mU/L and SD is 1.036 mU/L. We performed genome-wide analysis for inverse-normalized TSH on 20.7 million markers based on the linear mixed models using SAIGE[28] to account for sample relatedness. We used TSH batch (indicating whether TSH was measured in HUNT2 or HUNT3), genotyping batch, the first fifteen genetic principal components (PCs) (which account for 85% of the genetic variance among study samples), age at the measurement of TSH and sex as covariates. In a sensitivity analysis, we excluded HUNT participants with TSH outside the reference range (0.5–4.5 mU/L), and results were similar to the main analysis (Supplementary Data 15). Participation in HUNT is based on informed consent and the study has been approved by the Norwegian Data Protection Authority and the Regional Committee for Medical and Health Research Ethics in Central Norway.

**Michigan Genomics Initiative**. The MGI is a repository of electronic health and genetic data collected from patients at Michigan Medicine during pre-surgical encounters, who have consented to linking of genetic and clinical data for research purposes[26]. Participants were genotyped using Illumina Infinium CoreExome-24 bead arrays and genotype data were imputed to the HRC using the Michigan Imputation Server. Unrelated individuals with European ancestry were used for the GWAS on TSH levels. MGI participants (16,003) have at least one TSH measurement. Then, 5918 of these individuals with any thyroid disorders were excluded based on the ICD9 and ICD10 codes mapped to PheCodes[52] 193 (thyroid cancer), 244 (hypothyroidism), 245 (thyroiditis), and 246 (other disorders of thyroid), leaving 10,085 samples in the association analysis (53.4% are females). If more than one TSH measurement for an individual was available in the electronic health records, we used the average of the TSH levels for the individual in the analysis. The mean and SD for TSH in MGI are 1.914 and 1.175 mU/L, respectively. The mean age at TSH measurement is 55.90 years. We performed single-variant association testing of the inverse-normalized TSH levels on 17 million variants using a linear regression model as implemented in EPACTS with first four PCs, age, and sex as covariates. Later, we also performed single-variant association testing of thyroid cancer on the TSH-associated index variants using a logistic mixed model as implemented in SAIGE[28] with first 4 PCs, age and sex as covariates.

**ThyroidOmics**. The data set consists of a large meta-analysis for TSH performed by the ThyroidOmics consortium [http://www.thyroidomics.com] for up to 54,288 subjects of European ancestry[23]. Briefly, genotype data in 22 independent cohorts were imputed to 1000 Genomes, phase 1 version 3[60]. Eight million genetic markers were examined for association with inverse-normalized normal-range TSH levels in the meta-analysis including variants with MAF ≥ 0.5% and imputation score ≥ 0.4.

**UK Biobank**. The UKBB is a population-based cohort across the United Kingdom[27]. Non-sex-specific binary phenotypes (1283) have been constructed based on ICD9 and ICD10 codes mapped to PheCodes[28,56].

**FinnGen**. FinnGen is a public–private partnership project combining genotype data from Finnish biobanks and digital health record data from Finnish health registries (https://www.finngen.fi/en). Release 3 analysis contains 135,638 samples after quality control with population outliers excluded via PC analysis based on genetic data. Endpoints (1801) have been constructed from population registries and ICD10, and harmonizing definitions over ICD8 and ICD9.

**deCODE**. The data set contains 279,994 Icelandic participants of European ancestry at deCODE (1003 non-medullary thyroid cancers and 278,991 controls)[19]. The non-medullary thyroid cancer cases were identified from the ICR (http://www.krabbameinsskra.is/indexen.jsp?icd=C73). The mean age at diagnosis is 53 years for cases and the mean age of controls is 57 years[19].

**Columbus, USA**. The Columbus, USA, study contains 1580 patients with self-reported European ancestry histologically confirmed papillary or follicular thyroid carcinoma and 1628 controls[19]. The mean age at diagnosis is 43 years for cases and the mean age of controls is 45 years.

**Meta-analysis**. We performed fixed-effect meta-analysis based on inverse-variance weighting for HUNT, MGI, and ThyroidOmics in METAL[61]. Cochran's Q-test[62] for heterogeneity has been conducted using METAL[61]. We performed genomic control correction prior to and after the meta-analysis. We identified LD-independent genomic loci using LD-clumping based on association P-values from the meta-analysis using PLINK (–clump-r2 = 0.2 –clump-kb = 500 –clump-p1 = 5e-08 –clump-p2 = 5e-02). Overlapping loci were merged together. 1000 Genomes EUR[60] population was used as the reference panel for estimating LD.

**Stepwise conditional analysis**. We identified additional independent association signals at each locus using an approximate stepwise conditioning approach in GCTA-COJO[29] (--cojo-slct) and reported top variants with conditional P-values ≤ $5 \times 10^{-8}$.

**Gene-based association tests**. We performed gene-based SKAT-O tests using SAIGE-GENE[38,39] for 10,071 genes with at least two copies of missense and stop-gain alleles with MAF ≤ 0.5% and imputation quality score ≥ 0.8 in HUNT. We used the same covariates as the single-variant association tests: the first 15 PCs, age, sex, genotyping batch, and TSH measurement batch.

**Variant annotation and fine-mapping**. We annotated genetic variants using ANNOVAR[63]. We performed fine-mapping at all TSH loci using SuSiE[40], which estimated the number of causal variants and the 95% credible sets for each locus.

**Variance of TSH explained by loci**. We estimated the variance of TSH explained by TSH-associated loci as sum of effect size$^2 \times 2 \times$ MAF $\times (1 -$ MAF$)$ for all independent top hits, where effect size is in the unit of SD of TSH.

**Gene-enrichment tests**. We performed gene set and tissue enrichment analysis using DEPICT[47]. We included variants with association P-values $< 1 \times 10^{-5}$ based on the meta-analysis of HUNT, MGI, and ThyroidOmics in DEPICT. We used LD information from the European subsets of 1000 Genomes to construct loci within DEPICT.

**Phenome-wide association test**. For all 95 identified TSH index variants (in known or novel TSH loci; 4 out of 99 variants were excluded: rs1265091 and rs3104389 were HLA variants, and rs121908872 and 23:3612081 were not in UKBB), we carried out look-ups for their association with 1283 human diseases/conditioned constructed based on PheCodes mapped to ICD codes[27,28,48] and 274 quantitative traits (inverse ranked normalized, GWAS results from the Neale Lab) in the UKBB. We reported associations with P-value $< 5 \times 10^{-8}$.

**Polygenic score**. We computed PGS for TSH based on non-HLA significant index variants weighted by the effect size estimates from the meta-analysis of HUNT, MGI, and ThyroidOmics[23]. We excluded four index variants from the UKBB analysis because two were not available in UKBB (rs121908872 and 23:3612081) and two were located in the HLA region (rs1265091 and rs3104389). We constructed TSH PGS for 408,961 participants with White British ancestry in UKBB and tested for associations with 1283 binary phenotypes[27,28,48] using a logistic mixed model as implemented in SAIGE[28] with birth year, sex, and first four genetic PCs as covariates. We also constructed TSH PGS for 135,638 participants in FinnGen release 3 on the 87 non-HLA index variants (the HLA variants rs1265091 and rs3104389 were excluded from the analysis, 10 variants were not available in FinnGen r3: rs12027702, rs12138950, rs145153320, rs141751376, rs121908872, rs139352934, rs191633940, rs4571283, rs118039499, and 23:3612081) and tested for associations with 1801 binary phenotypes using a logistic mixed model as implemented in SAIGE[28] with age, sex, first ten genetic PCs and genotyping batch as covariates. In addition, we constructed TSH PGS for two previously reported thyroid cancer study populations and tested for associations with the risk of thyroid cancer: the Columbus, USA, study containing 1580 patients with self-reported European ancestry histologically confirmed papillary or follicular thyroid carcinoma and 1628 controls[19], and the Icelandic cohort with 1003 cases and 278,991 controls from deCODE[19]. In each data set, subjects were divided into quintiles according to their TSH PGS. OR (with 95% CI) of thyroid cancer was estimated for each quintile using the lowest quintile as the reference. The Nagelkerke's $r^2$ of TSH PGS for phenotypes in the UKBB was estimated as the

difference of $r^2$ in the full models with PRS and non-genetic covariates birth year, sex, and first four genetic PCs and $r^2$ in the reduced models with non-genetic covariates only without PRS using the R library rcompanion.

**Mendelian randomization**. We performed two-sample Mendelian randomization using summary data for 94 significant independent variants for TSH from the meta-analysis of HUNT and the ThyroidOmics consortium[23]. We excluded five SNPs from the analysis because they were not available in the thyroid cancer meta-analysis (rs121908872, rs4571283, and 23:3612081) or were located in the HLA region (rs1265091 and rs3104389). For the analysis, we calculated SNP-TSH effect estimates from the meta-analysis of HUNT and ThyroidOmics in up to 109,630 samples, excluding individuals with thyroid disorders. We calculated SNP-thyroid cancer effect estimates from the meta-analysis of UKBB[24], MGI, deCODE, and four other case–control data sets with European ancestry as reported in Gudmundsson et al.[19] (total number of cases/controls = 4146/731,903). We calculated SNP-goiter estimates from UKBB (1143 cases and 391,429 controls). We calculated the F-statistic to assess the strength of the instruments, where values >10 are indicative of adequate strength. We applied inverse-variance weighted, penalized weighted median, weighted median, weighted mode and pleiotropy residual sum and outlier methods using MR-Base[64] and MR-PRESSO[54]. The seven variants detected as outliers in the main analysis and excluded from the MR-PRESSO analysis were rs11156905, rs116909374, rs2993047, rs2928167, rs1479567, rs73234178, and rs925488.

**Materials**. Fetal bovine serum, ampicillin, and dithiothreitol (DTT) were from Sigma. QuikChange lightning site-directed mutagenesis kit was from Agilent. Plasmid purification kits were from Zymo Research. Protease inhibitor cocktail was from Roche. Penicillin/streptomycin, phosphate-buffered saline (PBS), and Dulbecco's modified Eagle's medium (DMEM) were from Gibco. Lipofectamine 2000 and subcloning efficiency DH5α-competent cells were from Invitrogen; LB broth base LB agar was from BD Difco.

**Mutagenesis of TG gene by PCR**. Utilizing the QuikChange Primer Design Program, we designed mutagenic primers (from Integrated DNA Technologies):
 RP.G67S, 5′-ccagcaagattgactatcattttggcattggactgtctgg-3′
 FP.G67S, 5′-ccagacagtccaatgccaaaatgatagtcaatcttgctgg-3′
 RP.P118L, 5′-ctgaatcctggcactgaagaaggtagagagcatcg-3′
 FP.P118L, 5′-cgatgctctctaccttcttcagtgccaggattcag-3′
 We incorporated PCR primers with the indicated point mutations using the QuickChange PCR mutagenesis kit as per the manufacturer's instructions (Agilent Technologies). We used the mTG cDNA sequence (NCBI database Reference Sequence: NM_009375.2) in the pcDNA3.1 expression vector as a template for primer design. We performed mutagenesis by PCR in a 2710 Thermal Cycler (Applied Biosystems) and confirmed mutations by DNA sequencing.

**Transient transfection and cell culture**. We cultured 293T cells in DMEM with 10% fetal bovine serum at 37 °C in a humidified 5% $CO_2$ incubator. We seeded 293T cells at 50,000 cells/well in 24-well plates 24 h prior to transfection and transiently transfected 500 ng of plasmid DNA per well using Lipofectamine 2000 transfection reagent according to the manufacturer's instructions. At 24 h post transfection, we washed the cells with PBS and re-fed with fresh serum-free media and incubated overnight. We collected bathing media and lysed the cells in RIPA buffer containing protease inhibitor cocktail.

**Western blotting**. We subjected samples to SDS-PAGE under reducing conditions, electrotransfer to nitrocellulose, and incubated in blocking buffer. We diluted primary antibody, rabbit polyclonal anti-TG 1:5000 in 5% bovine serum albumin/ TBS-T and incubated for 1 h at room temperature. We incubated anti-rabbit horseradish peroxidase-conjugated secondary antibody (1:5000 dilution in blocking buffer) for 30 min at room temperature. We visualized bands using the WesternBright Sirius kit (Advansta) and captured digital images (~5 s) in a Fotodyne workstation. We assessed differences in extracellular:intracellular mTg ratio (M/C, a value that is independent of transfection efficiency) using two-sample unpaired t-tests. $P$-value < 0.025 was declared significant with multiple testing taken into account, as two independent tests were conducted, one compared Tg-P118L variant to WT and the other one compared Tg-G67S to wild type. Full scans of western blottings are provided in Supplementary Fig. 14.

**Reporting summary**. Further information on research design is available in the Nature Research Reporting Summary linked to this article.

## Data availability
Data generated or analyzed during this study are available from the corresponding authors upon reasonable request, but access to individual-level information from the included studies may require approval from the individual studies and from the relevant research ethics committee due to privacy issues. Meta-analysis TSH summary statistics are available at http://csg.sph.umich.edu/willer/public/TSH2020/.

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

## Acknowledgements

We acknowledge the University of Michigan Precision Health Initiative and Medical School Central Biorepository for providing biospecimen storage, management, processing, and distribution services, and the Center for Statistical Genetics in the Department of Biostatistics at the School of Public Health for genotype data curation, imputation, and management in support of this research (for MGI cohort). The biochemical work at the University of Michigan was done with funding from NIH R01 DK-40344. This research has been conducted using the UK Biobank Resource under application number 24460. The HUNT Study is a collaboration between the HUNT Research Centre (Faculty of Medicine and Health Sciences, NTNU, Norwegian University of Science and Technology), Nord-Trøndelag County Council, Central Norway Regional Health Authority, and the Norwegian Institute of Public Health. The genotyping in HUNT was financed by the National Institutes of Health; University of Michigan; the Research Council of Norway; the Liaison Committee for Education, Research and Innovation in Central Norway; and the Joint Research Committee between St Olav's Hospital and the Faculty of Medicine and Health Sciences, NTNU. We thank Sean Caron for web development for data sharing. Support for this research was provided by NIH grant DK062370 (M.B.). Ohio State Cohort was supported by NIH P50CA168505 SPORE (M.D.R. and A.d.l.C.). Support for the HUNT and MGI analysis was additionally provided by R35HL135824 (C.J.W.) and R01HL109946 (C.J.W.). W.Z. was supported by the National Human Genome Research Institute of the National Institutes of Health under award number T32HG010464. G.D.S., B.B., H.R. and E.B.H. work in the Medical Research Council Integrative Epidemiology Unit at the University of Bristol MC_UU_00011/1.

## Author contributions

Study design: W.Z., B.B., O.K., P.A., K.H., C.J.W., and B.O.A. Analysis or interpretation of the results: W.Z., B.B., O.K., J.G., G.T., J.W., M.Z., J.B.N., L.C., M.M., A.T., S.N., S.S., U.T.S., A.C., J.K., M.K., M.O., P.T., L.G.F., S.E.G., B.N.W., W.O., H.R., E.B.H., I.S., G.D.S., A.P., T.R., W.E.H., J.G.J., L.S., S.L., M.D.R., L.X., L.A.K., H.H., R.T.N.-M., J.I.M., T.S.P., J.H., H.H., E.M.S., A.P., M.D., C.E.C., P.A., C.M.B., M.B., A.d.l.C., K.S., K.H., C.J.W., and B.O.A. Functional studies: O.K., C.E.C., and P.A. Drafting of manuscript: W.Z., B.B., O.K., S.E.G., B.N.W., W.O., I.S., P.A., M.B., C.J.W., and B.O.A. Statistical analysis of individual studies: J.G., G.T., J.W., M.Z., J.B.N., L.C., M.M., A.T., S.N., S.S., U.T.S., A.C., J.K., M.K., L.G.F., H.R., M.E.G., A.H.S., A.P., T.R., W.E.H., J.G.J, L.S., S.L., M.D.R., L.X., L.A.K., H.H., R.T.N.-M., J.I.M., T.S.P., J.H., H.H., E.M.S. A.P., M.D., C.M.B., M.B., A.C., and K.S. Critical revision of manuscript: all authors.

## Competing interests

The spouse of C.J.W. is employed at Regeneron Pharmaceuticals. The remaining authors declare no competing interests.

## Additional information

Wei Zhou [1,2,3,4,46]✉, Ben Brumpton [5,6,7,46], Omer Kabil[8,9,46], Julius Gudmundsson[10,46], Gudmar Thorleifsson[10,46], Josh Weinstock [11], Matthew Zawistowski [11], Jonas B. Nielsen [5,12,13], Layal Chaker[14,15,16], Marco Medici[14,15,16,17], Alexander Teumer [18,19], Silvia Naitza[20], Serena Sanna [20,21], Ulla T. Schultheiss[22,23], Anne Cappola[24], Juha Karjalainen[2,3,4,25], Mitja Kurki[2,3,4,25], Morgan Oneka [1], Peter Taylor[26], Lars G. Fritsche [11], Sarah E. Graham [12], Brooke N. Wolford [1,11], William Overton[11], Humaira Rasheed [5,6], Eirin B. Haug[5,6], Maiken E. Gabrielsen[5,27], Anne Heidi Skogholt[5,27], Ida Surakka[12], George Davey Smith [6,28], Anita Pandit[11], Tanmoy Roychowdhury[12], Whitney E. Hornsby[12], Jon G. Jonasson[29,30,31], Leigha Senter[32], Sandya Liyanarachchi[33], Matthew D. Ringel[34], Li Xu[35], Lambertus A. Kiemeney [36], Huiling He[33], Romana T. Netea-Maier [17], Jose I. Mayordomo[37], Theo S. Plantinga [38], Jon Hrafnkelsson[29], Hannes Hjartarson[29], Erich M. Sturgis[35], Aarno Palotie[2,3,4,25], Mark Daly[2,3,4,25], Cintia E. Citterio [9,39,40], Peter Arvan[9], Chad M. Brummett[41], Michael Boehnke [11], Albert de la Chapelle [33,47], Kari Stefansson [10,30,47], Kristian Hveem[5,42,43,47], Cristen J. Willer [1,12,44,47] & Bjørn Olav Åsvold [5,42,45,47]✉

[1]Department of Computational Medicine and Bioinformatics, University of Michigan, Ann Arbor, Michigan, USA. [2]Analytic and Translational Genetics Unit, Massachusetts General Hospital, Boston, Massachusetts, USA. [3]Program in Medical and Population Genetics, Broad Institute of Harvard and MIT, Cambridge, Massachusetts, USA. [4]Stanley Center for Psychiatric Research, Broad Institute of Harvard and MIT, Cambridge, Massachusetts, USA. [5]K.G. Jebsen Center for Genetic Epidemiology, Department of Public Health and Nursing, NTNU, Norwegian University of Science and Technology, Trondheim, Norway. [6]Medical Research Council (MRC) Integrative Epidemiology Unit, University of Bristol, Bristol, UK. [7]Department of Thoracic Medicine, St. Olavs Hospital, Trondheim University Hospital, Trondheim, Norway. [8]Department of Biological Chemistry, University of Michigan Medical School, Ann Arbor, Michigan, USA. [9]Division of Metabolism Endocrinology and Diabetes, University of Michigan Medical School, Ann Arbor, Michigan, USA. [10]deCODE genetics/AMGEN, 101 Reykjavik, Iceland. [11]Center for Statistical Genetics and Department of Biostatistics, University of Michigan School of Public Health, Ann Arbor, Michigan, USA. [12]Department of Internal Medicine, Division of Cardiology, University of Michigan Medical School, Ann Arbor, Michigan, USA. [13]Department of Epidemiology Research, Statens Serum Institute, Copenhagen, Denmark. [14]Erasmus MC Academic Center for Thyroid Diseases, Rotterdam, The Netherlands. [15]Department of Epidemiology, Erasmus Medical Center, Rotterdam, The Netherlands. [16]Department of Internal Medicine, Erasmus Medical Center, Rotterdam, The Netherlands. [17]Division of Endocrinology, Department of Internal Medicine, Radboud University Medical Centre, Radboud Institute for Molecular Life Sciences, 6500HB Nijmegen, The Netherlands. [18]Institute for Community Medicine, University Medicine Greifswald, Greifswald, Germany. [19]DZHK (German Center for Cardiovascular Research), Partner Site Greifswald, Greifswald, Germany. [20]Istituto di Ricerca Genetica e Biomedica, Consiglio Nazionale delle Ricerche Monserrato, Monserrato, Italy. [21]Department of Genetics, University of Groningen, University Medical Center Groningen, Groningen, The Netherlands. [22]Faculty of Medicine and Medical Center, Institute of Genetic Epidemiology, University of Freiburg, Freiburg, Germany. [23]Faculty of Medicine and Medical Center, Department of Medicine IV–Nephrology and Primary Care, University of Freiburg, Freiburg, Germany. [24]Division of Endocrinology, Diabetes, and Metabolism, University of Pennsylvania School of Medicine, Philadelphia, Pennsylvania, USA. [25]Institute for Molecular Medicine Finland, Helsinki Institute of Life Sciences, University of Helsinki, Helsinki 00014, Finland. [26]Thyroid Research Group, Systems Immunity Research Institute, Cardiff University School of Medicine, Cardiff, UK. [27]Faculty of Medicine and Health Sciences, Department of Public Health and Nursing, Norwegian University of Science and Technology, NTNU, Trondheim, Norway. [28]Department of Population Health Sciences, Bristol Medical School, University of Bristol, Bristol, UK. [29]Landspitali-University Hospital, 101 Reykjavik, Iceland. [30]Faculty of Medicine, University of Iceland, 101 Reykjavik, Iceland. [31]The Icelandic Cancer Registry, 105 Reykjavik, Iceland. [32]Division of Human Genetics, Ohio State University Comprehensive Cancer Center, Columbus, Ohio 43210, USA. [33]Department of Cancer Biology and Genetics, Ohio State University Comprehensive Cancer Center, Columbus, Ohio 43210, USA. [34]Division of Endocrinology, Diabetes, and Metabolism, The Ohio State University, Columbus, Ohio 43210, USA. [35]Department of Head and Neck Surgery, and Department of Epidemiology, The University of Texas MD Anderson Cancer Center, Houston, Texas 77030, USA. [36]Radboud University Medical Centre, Radboud

Institute for Health Sciences, 6500HB Nijmegen, The Netherlands. [37]University of Colorado Hospital, Aurora, Colorado 80045, USA. [38]Department of Pathology, Radboud University Medical Center, Radboud Institute for Molecular Life Sciences, 6500HB Nijmegen, The Netherlands. [39]Universidad de Buenos Aires, Facultad de Farmacia y Bioquímica, Departamento de Microbiología, Inmunología y Biotecnología/ Cátedra de Genética, Buenos Aires C1113AAD, Argentina. [40]CONICET-Universidad de Buenos Aires, Instituto de Inmunología, Genética y Metabolismo (INIGEM), C1120AAR Buenos Aires, Argentina. [41]Division of Pain Medicine, Department of Anesthesiology, University of Michigan Medical School, Ann Arbor, Michigan, USA. [42]HUNT Research Centre, Department of Public Health and Nursing, NTNU, Norwegian University of Science and Technology, Levanger 7600, Norway. [43]Department of Medicine, Levanger Hospital, Nord-Trøndelag Hospital Trust, Levanger 7600, Norway. [44]Department of Human Genetics, University of Michigan, Ann Arbor, Michigan, USA. [45]Department of Endocrinology, St. Olavs Hospital, Trondheim University Hospital, Trondheim, Norway. [46]These authors contributed equally: Wei Zhou, Ben Brumpton, Omer Kabil, Julius Gudmundsson, Gudmar Thorleifsson. [47]These authors jointly supervised this work: Albert de la Chapelle, Kari Stefansson, Kristian Hveem, Cristen J. Willer, Bjørn Olav Åsvold. ✉email: wzhou@broadinstitute.org; bjorn.o.asvold@ntnu.no

