## [Peer Review File · Nature Communications]

Reviewers' comments:

Reviewer #1 (Remarks to the Author):

Dr. Zhou and colleagues present a GWAS meta-analysis of TSH levels in the HUNT and MGI studies. They identify 17 novel loci, which are followed-up with fine-mapping and functional characterization. In addition, they perform a phenome-wide scan and a Mendelian randomization study to assess the causal effect of TSH levels on thyroid cancer. While the manuscript is easy to follow, there are some issues that I would like to bring to the Authors' attention.

- Distribution and characteristics of the two study samples (HUNG and MGI) should be clearly described at the beginning of the Results (age, sex, ancestry, other characteristics).

- I am confused by the GWAS design. Significance alpha for discovery is not defined. No criterion for replication is clearly stated. With HUNT and MGI together, the Authors had at their disposal >65K GWAS samples. The Authors could have followed a discovery-replication approach, using ThyroidOmics data as replication (testing 1-sided replication penalized by the number of multiple tests) or a joint analysis of HUNT and MGI and ThyroidOmics, which would have resulted in a more powerful single-stage analysis. Instead, the Authors' choice was half-a-way between the two solutions: they used HUNT+MIG for discovery, selected SNPs with p-value < 5e-06 and, on these, perform a meta-analysis with ThyroidOmics. The Authors claim 17 novel loci but, if all data were pooled in a single analysis encompassing 65K + 54K samples, the yield would have been probably much bigger. The rationale for such a choice, which looks like a missed opportunity, looks unclear. Please, provide convincing explanation or review the work following a more standard design.

- On page 4, for several variants, MAF and effect size are given only for the HUNT study and not for the meta-analysis of Hunt and MGI (or Hunt-MGI-ThyroidOmics). Why? In addition, what does it mean that rs139352934) is 22 times more frequent in HUNT than in other non-Finnish Europeans? Is it a Scandinavian variant then?

- In Table 1, directions of the effects for each of the studies included in the ThyroidOmics meta-analysis is probably also available. It would be very informative to add those effect directions as well, and compute an overall heterogeneity index that accounts for all ThyroidOmics studies, HUNT, and MGI.

- Given the large sample size of the HUNT study, small differences in the laboratory method to assess TSH levels might affect association estimated, especially because laboratory methods are not randomized across participants (PMID 28374192). Did the Authors control for laboratory method in their GWAS? If not, an analysis should be performed to guarantee that no spurious stratification has arisen.

- In both HUNT and MGI studies: what was the reason to adjust for just 4 PCs? They two studies have very different setting and genetic structure. The choice of exactly the same number of PCs sounds curious.

- Is it correct that the Authors performed an inverse-variance based meta-analysis of HUNT, MGI and ThyroidOmics GWAS results? While in MGI and ThyroidOmics, the inverse normal transformation of TSH was analyzed, in the HUNT study, TSH was not transformed. Probably a meta-analysis of p-values or z-scores weighted by sample size would be more appropriate.

- The MGI is a study on pre-surgical individuals. It is a special study, as reflected, for instance, by the large proportion of individuals affected by any thyroid disease who have been excluded from the analysis. In their Mendelian randomization analyses, did the Authors consider the problem of different population background between exposure and outcome studies? In other words: in MR

studies, it is assumed that exposure and outcome refer to a similar context (similar age, population characteristics, etc.). Has this been verified? How?

- Would it make sense to perform bidirectional MR analysis to test whether there might be a causal effect of thyroid cancer on TSH levels?

- In Fig. 1, please avoid barplots and use boxplots instead, showing the actual points. In the figure and in the text, please, define the number of independent statistical units and how you took multiple testing into account. Please, avoid p-value dichotomization using "*", and discuss the results in the light of the underlying statistical assumptions.

- Fig 2: please, provide confidence interval for prevalence as well. In addition, use more appropriate ways to estimate confidence intervals for the odds ratios, as they cannot be >1. Finally, discuss the very different prevalence in Columbus.

- I don't think ThyroidOmics can be defined a "cohort", as it was itself a consortium including many studies.

- When describing the methods for the HUNT study and for the Phenome-wide scan, please, indicate which version of ICD codes was used (9, 10, 11?).

- When describing the methods for the HUNT meta-analysis, I assume "the first four principal components" should read "the first four genetic principal components".

- There are several typos / misprints throughout the document. Please, read carefully and fix. There are unexplained acronyms (eg: HRC)

Reviewer #2 (Remarks to the Author):

Zhou et al. conducted a GWAS on TSH levels using data from HUNT study and the MGI biobank. In the replication phase using data from ThyroidOmics, 58 loci were replicated of which 17 were novel. In order to identify causal variants, fine-mapping analysis were performed and functional follow-up with a focus on the significant locus located in the TG gene.

The authors also analyzed the pleiotropic effect of the top 85 TSH variants (not located in HLA region) with 1,283 human diseases and 274 continuous traits using the data from the UK Biobank and they identified 17 variants associated to 29 diseases and 29 variants associated to at least one quantitative traits. Analysis of the pleiotropic effects of these 85 variants with thyroid cancer risk was carried on using results from the meta-analysis of UK Biobank and deCODE data; identifying 17 variants associated to both TSH levels and thyroid cancer risk.

A TSH polygenic risk score was derived from the 85 variants highlighted by the meta-analysis of GWAS (HUNT study, MGI biobank and ThyroidOmics consortium) and a pheWAS analysis was conducted using data from the UK Biobank dataset (1283 diseases) and from FinnGen, showing a positive association with hypothyroidism risk and an inverse association with risk of goiter in both studies. An inverse association between the TSH PGS and thyroid cancer risk was also reported in the UK biobank, and replicated in the Columbus, deCODE and FinnGen studies.

At last, in order to evaluate the association between TSH levels and thyroid cancer risk using a two-sample mendelian randomization approach, an instrumental variable of TSH levels was derived from the 83 significant variants from the meta-analysis of HUNT and the ThyroidOmics consortium and its association with TC risk was tested in the meta-analysis of UKBB, MGI and deCODE. The association between the instrumental variable for TSH levels was also tested in association with goiter risk using the UKBB data. These analyses suggested a protective effect of

high TSH levels on the risk of thyroid cancer and goiter.

This is a large GWAS study on TSH levels with analysis of its pleiotropic effects with thyroid conditions and disorders. The results are interesting, however I think some major information were missing making the article difficult to understand.

Here are some major comments:

1- Only data from HUNT and MGI are described. Description of deCODE, UK BB, FinnGen, Columbus and ThyroidOmics studies are missing. In particular, it would be important to describe how the phenotypes of interest (thyroid cancer, TSH levels, and thyroid disorders) were defined and how many individuals for each phenotype were available in these studies. If only summary statistics were available for these studies, this need to be mentioned.

2- A descriptive table of the different studies with the number of cases for the phenotypes of interest and some indication on the age group and sex distribution would be helpful. Especially as age and sex have important effects on thyroid disorders.

3- If definition of thyroid conditions differs across the studies (self-declared vs validated cases, information on histology of thyroid cancer,...) this need to be commented further as limits of the study in the discussion part. It is known that information on self-declared thyroid cancer can be very poor in some cohort studies, and also risk factors of medullary thyroid cancer are really different from papillary or vesicular thyroid cancer.

4- In the TSH GWAS, data from HUNT and MGI studies were used. When several measurements of TSH levels were available for an individual, it is mentioned that in HUNT study, the last measurement was used, while in MGI study the authors used the average of the TSH levels. Why the authors did not used the same rule for both studies, this need to be justified/commented?

5- In general, the tables, supplementary tables and figures titles are not explicit and there not enough explanation in the legend. Abbreviations in tables should be clarified and meaning of some columns are not clear. For instance, I don't understand the column "direction" in Table 1.

6- The authors should explain the principle of the SuSiE method used for fine-mapping. Also, it is unclear how to interpret the results shown in Supplementary Table 4. A footnote with more explanation is required. Also, only 51 loci appeared in this table while 58 loci were highlighted by the GWAS. The authors should explain this difference.

7- The authors should explain why they focused on TG genes and not on other loci while they conducted the fine-mapping analysis.

8- In page 6, the authors explained why that they limited the analyses of pleiotropic effect on the 85 non HLA variants in the paragraph "Associations of PGS of TSH with other phenotypes". This explanation should appear in the previous paragraph "Pleiotropic effects of TSH loci". Also, it is not clear how many variants were removed from the analysis because they were located in the HLA region.

9- It seems that the authors are confused with the difference between case-control and cohort studies while citing the Columbus cohort as a "case-control cohort" (on page 7).

It does not make sense to look at the thyroid prevalence using the data from a case-control study. I do not understand the rationale to show both prevalence and OR of thyroid cancer in figure 2. If logistic regression were not adjusted on any factors, then both results (right and left panel) are similar but not on the same scale. Also, the results on prevalence are difficult to compare between studies. I suggest to remove the results on prevalence.

10- The authors should give the weight of the SNPs that permit to construct the instrumental variable for TSH levels, as it seems that it is derived from different studies (HUNT and the ThyroidOmics consortium) than for the polygenic score (HUNT study, MGI biobank and ThyroidOmics).

Also, they should explain why they did not use the same weight between the instrumental variable for TSH levels and the polygenic score. Also why FinnGen study was used to replicate the association between PGS and goiter but was not used for the MR analysis on goiter?

11- 17 pleiotropic variants were reported between TSH levels and thyroid cancer risk. To my understanding, the instrumental variable included those variants in the mendelian randomisation (MR) analysis. As the presence of pleiotropic effect violates the condition of application of MR, a sensitivity analysis excluding these SNPs may be useful.

12- In the discussion part, the authors should discuss the results of the TSH GWAS in regards to the results from previous GWAS. Is there any loci that were previously highlighted that were not replicated in this new GWAS?

13- In page 8, it is stated that the causal direction between TSH levels and thyroid cancer risk is unexpected. I think the author should mention and discuss recent findings from epidemiological studies that showed an inverse association between TSH levels measures and papillary thyroid cancer. Here are some references:

-Franceschi S, Rinaldi S. TSH, Thyroid Hormone, and PTC-Letter. *Cancer Epidemiol Biomarkers Prev*. 2018 Feb;27(2):227.

- Huang H, Rusiecki J, Zhao N, Chen Y, Ma S, Yu H, et al. Thyroid-stimulating hormone, thyroid hormones, and risk of papillary thyroid cancer: a nested case-control study. *Cancer Epidemiol Biomarkers Prev* 2017;26:1209–18.

- Rinaldi S, Plummer M, Biessy C, Tsilidis KK, Ostergaard JN, Overvad K, et al. Thyroid-stimulating hormone, thyroglobulin, and thyroid hormones and risk of differentiated thyroid carcinoma: the EPIC study. *J Natl Cancer Inst* 2014.

Minor comments

1- Some studies were named differently along the manuscript. The Columbus study was named Ohio cohort in introduction, while deCODE study is sometimes named Iceland study

2- I think that the description of the GWAS analysis should not be in the studies description but should appear separately to make it clearer.

3- The sentence "We performed genomic control correction was performed on HUNT and MGI..." (page 10) need to be corrected

4- Page 11: the "materials" section need to be rephrase

5- Order of table need to be revised, for instance Supp Table 13 is cited after Supp Table 14 in the text

Reviewer #3 (Remarks to the Author):

This study identifies 58 genome-wide significant loci for TSH levels, 17 of which are novel. These loci show pleiotropic effects on various diseases and traits, and provide genetic evidence for a causal relationship between TSH levels and the risk of thyroid cancer and goiter. Associations in the thyroglobulin (TG) locus are fine-mapped to two missense variants in TG, and functional work provides evidence for their role in thyroglobulin secretion.

This interesting study contains several novel findings that significantly expands the current genetic knowledge in this field. However, I have several comments/concerns presented below in the same order as presented in the manuscript (from Results to the Methods section):

Was there any evidence of inflation in the GWAS statistics due to reasons other than polygenicity (e.g. population structure)? It would be informative to investigate this using the univariate LD score regression intercept.

Four rare variants (rs546738875, rs145153320, rs139352934, and rs121908872) that were associated with TSH levels in HUNT were not available in the ThyroidOmics dataset for replication. The authors explain that the frequency of the risk allele were somewhere between 11-22 times higher in HUNT than the other studies. However, how are the authors convinced that these results are not false positive findings in HUNT? In addition, were these variants genome-wide significant in the original analysis or only after performing the conditional analysis? If the latter, could this be an artifact due to the conditional analysis?

Were the significantly associated rare variants as well as those included in the SKAT-O analysis imputed? If so, was their imputation quality reasonably high? Please provide their imputation quality scores in a supplementary Table.

The authors mention: "An intronic variant (rs6544660) in the thyroid adenoma associated gene THADA was recently identified to be associated with TSH". However, this variant has been reported as a novel discovery in this study in Table 1.

Were there any sample overlaps between the discovery and target datasets used for the PGS and MR analyses? I noticed that this study by design attempted to avoid such sample overlaps; however, it would be ideal to ensure that the results are not biased due to hidden sample overlaps or relatedness between individuals in the discovery and target datasets. This can be investigated, for instance, using the bivariate LD score regression intercept.

Apart from thyroid cancer and goiter, did the authors investigate causality of the other phenotypes that were associated with TSH PGS? Performing MR for the traits that were significantly associated in PGS analyses may help understand whether these genetic links are consistent with pleiotropy or causality. I do not have a strong preference whether all these analyses should be included in this paper; however, considering the PGS results, some reader may be interested to know whether the genetically predicted TSH levels are causally associated with any of the other traits that came up in the PGS analyses.

For the MR findings for thyroid cancer, although the CIs of the MR estimates overlap between the sensitivity tests, most of the test statistics are not significant (e.g. MR Egger, simple mode, and weighted mode in Suppl Table 15). Please discuss this in the text. Is this a power issue?

It would be helpful to discuss briefly, what would be the clinical implication of showing evidence of causality of higher TSH levels on reduced risk of thyroid cancer? Is there any evidence showing that increasing TSH levels is capable of treating cancer? Alternatively, in terms of prevention strategies, is increasing TSH levels practical, or can this lead to thyroid dysfunction and adverse metabolic consequences?

How well did the TSH PGS predict the phenotypes investigated? Providing r^2 or AUC in a supplementary table can be informative. Although I would expect that, similar to the other complex traits, a good prediction accuracy requires much larger sample sizes.

The authors mention that the previously identified 47 independent TSH-associated loci account for 14% of TSH heritability. How much do the novel loci add to the explained heritability? In addition, based on the other complex traits, explaining 14% of variance in heritability from only 47 SNPs seems to be too high. Is that because several common SNPs with large effect sizes are involved? Alternatively, please ensure that the explained heritability is estimated accurately.

For the significant traits in PheWAS (e.g. for the 29 disease phenotypes), it would be interesting to know whether the genetic link with those traits also expand throughout the genome. The overall genetic correlation can be investigated using summary statistics using the LD score regression approach.

Although having thyroid gland as the most significant tissue in the DEPICT tissue enrichment analysis is expected and interesting, none of the tissue enrichment associations were statistically significant after correcting for multiple testing. I suggest specifying this in the text.

To get an unbiased understanding of genetic link between TSH and thyroid cancer, instead of using only SNPs that were also significant for thyroid cancer, it was better to include all the significant TSH SNPs in the analysis presented in supplementary Fig 6 as well as binomial tests for the direction of effect. In addition, adding correlation coefficient or variance explained to the supplementary Fig 6 would be helpful. Having said that, the caveat for using all the TSH significant SNPs in the binomial tests for the direction of effect would be that some SNPs might show unexpected direction of effect for thyroid cancer while their CIs overlap with the expected direction.

The authors have inversed normalized TSH levels for each study separately, and report the association results on the SD scale per study. Was the SD similar across all the studies? Different SD across studies makes it difficult to convert the meta-analyzed effect estimates back on the actual TSH levels scale.

The investigators specify that they have used different methods for measuring TSH levels across studies. Does this bias the results? Do they get similar results after adding the measurement method as a covariate in their regression analysis?

The authors mention that "We computed polygenic score (PGS) for TSH based on the 84 non-HLA significant index variants weighted by the effect size estimates based on the meta-analysis results of the HUNT Study and the MGI Biobank and then with the GWAS results from the ThyroidOmics consortium". Does this mean that PGS was calculated from the HUNT + MGI + ThyroidOmics meta-analysis, or was PGS based on ThyroidOmics estimates calculated separately for a comparison?

For HUNT, the authors have noted that if multiple TSH measurements were available, they used the most recent one. However, for MGI they used the average of the TSH levels in the same scenario. Was this inconsistency justified, or is it due to different analysis plans per study prior to conducting the meta-analysis?

It was not clear why the authors have only included significant variants from the meta-analysis of HUNT + MGI in their DEPICT analysis. Would the final meta-analyzed results including ThyroidOmics dataset be more powerful for their DEPICT analysis?

Please provide details of the stepwise conditional analysis in the Methods section. Was this done using GWAS summary statistics or raw genotypes?

The authors say: "For all genetic variants with P-value < 5×10^{-6} , we conducted another meta-analysis with up to 54,288 additional samples in ThyroidOmics". Why the meta-analysis was not performed using all the SNPs in these three studies? Was this because of access restrictions to ThyroidOmics data?

Response to Reviewers Comments

Title: GWAS of thyroid stimulating hormone highlights its pleiotropic effects and genetic association with low thyroid cancer risk

We thank the reviewers and editors for their thoughtful and constructive comments that helped improve the manuscript. Below are our detailed responses (in bold). Quoted text from the manuscript is highlighted in blue.

Editor's comments:

For Figure 1, please state how you controlled for transfection efficiencies between wt and mutant plasmids (i.e. did you include GFP or a similar control to make sure that the differences in expression/secretion are not because transfection was less efficient for the mutants?). Why was a t test used and not ANOVA for comparing multiple groups? Please normalize the Western blot data with a housekeeping gene that was detected in the same sample preparation (at least for the cell lysates).

Thank you for the query, and we apologize for any confusion in our previous writing. For each construct, we measured secretion efficiency in the 24 - 48 h time window post-transfection. As an aside, using GFP-tagged constructs in our protocol, we have found transfection efficiency to exceed 80% — but the essential point in our analysis is that secretion efficiency is internally controlled. For each sample, what is measured is the relative protein level recovered from the cells versus that recovered from the medium bathing those same cells. These internally controlled results (analyzed from identical fractions loaded from cells seeded and transfected in parallel at identical cell density) do not change by normalization to transfection efficiency or housekeeping gene expression.

To clarify, we have modified the text in the subsection Functional follow-up of missense variants in the thyroglobulin gene (*TG*) of Results "Then, we examined the intracellular versus secreted levels of the mTg-WT and these two human Tg variants (Tg-p.P118L and Tg-p.G67S). Transfected cells were incubated overnight and the culture medium and cell lysates were analyzed by SDS-PAGE and immunoblotting with anti-Tg antibody. The experiment was independently repeated three times and the results analyzed in a manner that is independent of transfection efficiency." We have revised the last sentence in the subsection Western blotting of Methods, "We assessed differences in extracellular:intracellular mTg ratio (M/C, a value that is independent of transfection efficiency) using two-sample unpaired t-tests. P-value < 0.025 was declared significant with multiple testing taken into account, as two independent tests were conducted, one compared Tg-P118L variant to wild type and the other one compared Tg-G67S to wild type."

The second part of the question was about the use of the t-test to compare two groups (each variant versus WT). T-test was used to assess whether either variant exhibited a secretion efficiency that was statistically different from WT. We believe this is the appropriate analysis, and we have removed the second last sentence in the subsection Functional follow-up of missense variants in the thyroglobulin gene (*TG*) of Results to avoid extraneous statistical claims from the manuscript, "~~However, the deleterious effect of G67S on the secreted levels of Tg was not as strong as the effect of P118L (P-value for the difference is 0.018).~~"

Please be aware that for certain types of new data, including most types of genetic data, journal policy is that deposition in a community-endorsed, public repository is generally mandatory prior to publication. Data submission can be a lengthy process, and we strongly suggest that you begin this well in advance of potential publication to avoid delays later on. Please include a statement about data availability (specifically about the sharing of GWAS summary statistics) in your point-by-point letter accompanying your revisions.

Thank you for the instruction. We have included the Data availability section in the manuscript.

“Data generated or analyzed during this study are available from the corresponding authors upon reasonable request. Meta-analysis TSH summary statistics are available at: <http://csg.sph.umich.edu/willer/public/TSH2020/>.”

Meta-analysis summary statistics for TSH will be available at the link above upon acceptance of the paper.

Please see decision letter above for more detailed information about data requirements and policy. If you are unable to make your data publicly available for exceptional reasons, please get in touch with me now to discuss this further.

Reviewers' comments:

Reviewer #1 (Remarks to the Author):

Dr. Zhou and colleagues present a GWAS meta-analysis of TSH levels in the HUNT and MGI studies. They identify 17 novel loci, which are followed-up with fine-mapping and functional characterization. In addition, they perform a phenome-wide scan and a Mendelian randomization study to assess the causal effect of TSH levels on thyroid cancer. While the manuscript is easy to follow, there are some issues that I would like to bring to the Authors' attention.

1. Distribution and characteristics of the two study samples (HUNT and MGI) should be clearly described at the beginning of the Results (age, sex, ancestry, other characteristics).

We thank the reviewer for pointing this out. We have described the sample characteristics for HUNT MGI and ThyroidOmics in the subsection Description of data sets.

In the paragraph for HUNT “TSH measurements were available for 55,342 individuals who were genotyped (54.4% are females), after excluding those with thyroid disorders based on self-report and cancer registry data (7.34%). If TSH was measured in both HUNT2 and HUNT3, the earlier HUNT2 measure was used. The mean age at TSH measurement is 51.79 years. The mean TSH in HUNT is 1.659 mU/L and SD is 1.036 mU/L.”

In the paragraph for MGI “16,003 MGI participants have at least one TSH measurement. 5,918 of these individuals with any thyroid disorders were excluded, leaving 10,085 samples in the association analysis (53.4% are females). If more than one TSH measurement for an individual was available in the electronic health records, we used the average of the TSH levels for the individual in the analysis. The mean and SD for TSH in MGI are 1.914 mU/L and 1.175 mU/L, respectively. The mean age at TSH measurement is 55.90 years.”

In the paragraph for ThyroidOmics “The data set consists of a large meta-analysis for TSH performed by the ThyroidOmics consortium [<http://www.thyroidomics.com>] for up to 54,288 subjects of European ancestry. Briefly, genotype data in 22 independent cohorts were imputed to 1000 Genomes¹, phase 1 version 3. Eight million genetic markers were examined for association with inverse-normalized normal-range TSH levels in the meta-analysis including variants with MAF \geq 0.5% and imputation score \geq 0.4.”

2. I am confused by the GWAS design. Significance alpha for discovery is not defined. No criterion for replication is clearly stated. With HUNT and MGI together, the Authors had at their disposal >65K GWAS samples. The Authors could have followed a discovery-replication approach, using ThyroidOmics data as replication (testing 1-sided replication penalized by the number of multiple tests) or a joint analysis of HUNT and MGI and ThyroidOmics, which would have resulted in a more powerful single-stage analysis. Instead, the Authors' choice was half-a-way between the two solutions: they used HUNT+MIG for discovery, selected SNPs with p-value < 5e-06 and, on these, perform a meta-analysis with ThyroidOmics. The Authors claim 17 novel loci but, if all data were pooled in a single analysis encompassing 65K + 54K samples, the yield would have been probably much bigger. The rationale for such a choice, which looks like a missed opportunity, looks unclear. Please, provide convincing explanation or review the work following a more standard design.

We apologize for the unclear description of our study design. We agree with the reviewer that pooling HUNT, MGI and ThyroidOmics together in a single analysis would lead to higher power for discovery. Thus, we have now updated our analysis by meta-analyzing HUNT (N=55,342), MGI (N = 10,085), and ThyroidOmics together (up to N=54,288). This new meta-analysis for TSH levels has increased the number of genome-wide significant loci from 58 to 74 and number of novel loci for TSH from 17 to 28. We have also updated the whole manuscript with the new results.

3. On page 4, for several variants, MAF and effect size are given only for the HUNT study and not for the meta-analysis of Hunt and MGI (or Hunt-MGI-ThyroidOmics). Why? In addition, what does it mean that rs139352934 is 22 times more frequent in HUNT than in other non-Finnish Europeans? Is it a Scandinavian variant then?

Again, we thank the reviewer for pointing this out. As the missense variants in the gene *B4GALNT3* and *TSHR* that we highlighted on page 4 are only observed in the HUNT study, MAF and effect sizes are given only for HUNT. Moreover, the effect size and MAF of the top variants at significant TSH loci are included in Table 1 and Supplementary Table 1 and 2. rs139352934 has frequency 0.2% in HUNT. In [gnomAD \(v2, \[https://gnomad.broadinstitute.org/variant/14-81610228-G-A?dataset=gnomad_r2_1\]\(https://gnomad.broadinstitute.org/variant/14-81610228-G-A?dataset=gnomad_r2_1\)\)](https://gnomad.broadinstitute.org/variant/14-81610228-G-A?dataset=gnomad_r2_1), it is rare in Finnish (0.008%) and non-Finnish Europeans (0.009%). Although the variant has higher frequency in Swedish (0.015%) than in Finnish and non-Finnish Europeans, it is not clear whether it is a Scandinavian variant. We have discussed this in the first paragraph of the Discussion section “Additional independent signals were identified among several loci based on GWAS results in the meta-analysis and LD information in the HUNT study, including two rare variants rs546738875 and rs145153320 at the *B4GALNT3* locus and two rare missense variants *TSHR* p.A553T (rs121908872) and *TSHR* p.R609Q (rs139352934), which have been observed to be associated with congenital hypothyroidism in previous family studies^{2,3}. *TSHR* p.R609Q (rs139352934) is the most strongly associated with TSH in the thyroid stimulating hormone receptor gene *TSHR* with an effect size greater than one standard deviation of TSH (1.036 mU/L). As these rare variants were only imputed in HUNT, not in MGI or ThyroidOmics, further follow-up to verify the associations is needed.”

4. In Table 1, directions of the effects for each of the studies included in the ThyroidOmics meta-analysis is probably also available. It would be very informative to add those effect directions as well, and compute an overall heterogeneity index that accounts for all ThyroidOmics studies, HUNT, and MGI.

We thank for the reviewer's suggestion. The directions of the effects for each of the studies and the heterogeneity P-values have been included in Table 1 for top variants at novel loci and in the Supplementary Table 2 for top variants at known loci.

5. Given the large sample size of the HUNT study, small differences in the laboratory method to assess TSH levels might affect association estimated, especially because laboratory methods are not randomized across participants (PMID 28374192). Did the Authors control for laboratory method in their GWAS? If not, an analysis should be performed to guarantee that no spurious stratification has arisen.

In the updated GWAS, we have now controlled for laboratory method although the two measurement methods for TSH in HUNT2 and HUNT3 have been previously shown to yield similar measurements (PMID: 23975540, our reference #54). We have used a covariate "TSH batch" to indicate whether the TSH was measured in HUNT2 or HUNT3 in the GWAS and included the description of the GWAS model that has been used in HUNT in the Description of data sets subsection in the METHODS section "We used TSH batch (indicating whether TSH was measured in HUNT2 or HUNT3), genotype batch, the first fifteen genetic principal components (PCs), which account for 85% of the genetic variance among study samples, age at the measurement of TSH and sex as covariates."

6. In both HUNT and MGI studies: what was the reason to adjust for just 4 PCs? They two studies have very different setting and genetic structures. The choice of exactly the same number of PCs sounds curious.

In the current (updated) GWAS for HUNT, we used the first 15 PCs, which accounts for 85% of genetic variance among HUNT samples. Similarly, in the MGI study, 4 PCs were used as they are sufficient to remove any residual ancestry bias in the MGI cohort.

7. Is it correct that the Authors performed an inverse-variance based meta-analysis of HUNT, MGI and ThyroidOmics GWAS results? While in MGI and ThyroidOmics, the inverse normal transformation of TSH was analyzed, in the HUNT study, TSH was not transformed. Probably a meta-analysis of p-values or z-scores weighted by sample size would be more appropriate.

We apologize for the unclear description. The inverse normal transformation of TSH was also conducted in the HUNT study for GWAS. We have included the description in the Description of data sets subsection in the Methods section "We performed genome-wide association analysis for inverse-normalized TSH on 20.7 million markers based on the linear mixed models using SAIGE to account for sample relatedness."

8. The MGI is a study on pre-surgical individuals. It is a special study, as reflected, for instance, by the large proportion of individuals affected by any thyroid disease who have been excluded from the analysis. In their Mendelian randomization analyses, did the Authors consider the problem of different population background between exposure and outcome studies? In other words: in MR studies, it is assumed that exposure and outcome refer to a similar context (similar age, population characteristics, etc.). Has this been verified? How?

In two-sample MR studies, the exposure and outcome study populations do not necessarily refer to a similar context (typically the study contexts differ, with outcome information often obtained from large case-control studies enriched for the outcome), but it is assumed that the effects of the genetic instrument on the exposure should be the same in the outcome as in the exposure study. We have ensured that both the exposure and outcome associations are obtained from European ancestry populations. Further, to explore if the causal effects could be sensitive to age differences (i.e. restricted to thyroid pathophysiological changes occurring in old age), we repeated the MR analysis using SNP-TSH effect estimates obtained among those younger than 50 years of age at the time of TSH measurement, and we observed results similar to the main analysis as shown in the Supplementary Table 18.

9. Would it make sense to perform bidirectional MR analysis to test whether there might be a causal effect of thyroid cancer on TSH levels?

A possible causal effect of thyroid cancer on TSH levels would be difficult to examine in this setting. Although we could possibly examine the effect of a genetic liability to thyroid cancer on TSH levels, this could not be translated to the effect on TSH levels once thyroid cancer has developed. Further, detection of thyroid cancer would typically lead to thyroidectomy. This would, if untreated, lead to severe hypothyroidism with severely elevated TSH levels, whereas the levothyroxine treatment given to all thyroidectomized patients would lead to normalized or suppressed TSH levels, depending on the dose of levothyroxine given.

10. In Fig. 1, please avoid barplots and use boxplots instead, showing the actual points. In the figure and in the text, please, define the number of independent statistical units and how you took multiple testing into account. Please, avoid p-value dichotomization using "*", and discuss the results in the light of the underlying statistical assumptions.

We have re-plotted Figure 1 using boxplots overlaid with the actual points. We have removed the "*" to indicate P-values. In the Western blotting subsection in the Methods section, we have described the significant threshold for P-values "We assessed differences in extracellular:intracellular mTg ratio (M/C, a value that is independent of transfection efficiency) using two-sample unpaired t-tests. $P < 0.025$ was declared significant with multiple testing taken into account, as two independent tests were conducted, one compared Tg-P118L variant to wild type and the other one compared Tg-G67S to wild type."

At the end of the subsection Functional follow-up of missense variants in the thyroglobulin gene (TG) of the Results section, we have discussed the results "Compared to the wild type, the Tg-P118L variant showed a significant reduction in the M/C ratio 0.6:1 (P-value = 0.0051) and the Tg-G67S variant also showed a significant reduction in the M/C ratio 1.96:1 (P-value = 0.0095)."

11. Fig 2: please, provide confidence interval for prevalence as well. In addition, use more appropriate ways to estimate confidence intervals for the odds ratios, as they cannot be >1 . Finally, discuss the very different prevalence in Columbus.

We have provided confidence intervals for prevalence in Figure 2. The odds ratios and corresponding confidence intervals (unlike the prevalence / probability) can be greater than 1. Because the Columbus, USA study is a case-control study, its study population has a much higher prevalence than

the other three data sets. To avoid confusion, we have moved the plots for the Columbus, USA study to Supplementary Figure 10.

12. I don't think ThyroidOmics can be defined a "cohort", as it was itself a consortium including many studies.

We apologize for the confusion. The ThyroidOmics is a consortium (<http://www.thyroidomics.com>). The TSH meta-analysis in the ThyroidOmics is based on 22 independent cohorts (Teumer et.al. 2018). We have replaced cohort with consortium in our manuscript.

13. When describing the methods for the HUNT study and for the Phenome-wide scan, please, indicate which version of ICD codes was used (9, 10, 11?).

Thanks for this question. The phenome-wide scan in the UK Biobank was conducted using phenotypes that were defined using the ICD9 and ICD10 code-based phecode map and were previously reported in our phenome-wide GWAS scan for the UK Biobank (PMID 30104761, our reference #26, PMID 31553307, our reference 44, PMID 24270849, our reference #51).

We have described this in the Descriptions of data sets subsection of the Methods section:

“UK Biobank

The UK Biobank (UKBB) is a population-based cohort across the United Kingdom⁴. 1,283 non-sex specific binary phenotypes have been previously constructed based ICD9 and ICD10 codes mapped to phecodes^{5,6}.

HUNT

TSH measurements were available for 55,342 individuals who were genotyped (54.4% are females), after excluding those with thyroid disorders based on self-report and cancer registry data (7.34%).”

14. When describing the methods for the HUNT meta-analysis, I assume "the first four principal components" should read "the first four genetic principal components".

Yes, thanks for pointing this out. We have added “genetic” to in the Description of data sets subsection in the Methods section.

15. There are several typos / misprints throughout the document. Please, read carefully and fix. There are unexplained acronyms (eg: HRC)

We apologize for the typos and unexplained acronyms, which have now been corrected and edited accordingly.

Reviewer #2 (Remarks to the Author):

Zhou et al. conducted a GWAS on TSH levels using data from HUNT study and the MGI biobank. In the replication phase using data from THyroidOmics, 58 loci were replicated of which 17 were novel. In order to identify causal variants, fine-mapping analysis were performed and functional follow-up with a focus on the significant locus located in the TG gene.

The authors also analyzed the pleiotropic effect of the top 85 TSH variants (not located in HLA region) with 1,283 human diseases and 274 continuous traits using the data from the UK Biobank and they identified 17 variants associated to 29 diseases and 29 variants associated to at least one quantitative traits. Analysis of the pleiotropic effects of these 85 variants with thyroid cancer risk was carried on using results from the meta-analysis of UK Biobank and deCODE data; identifying 17 variants associated to both TSH levels and thyroid cancer risk.

A TSH polygenic risk score was derived from the 85 variants highlighted by the meta-analysis of GWAS (HUNT study, MGI biobank and ThyroidOmics consortium) and a pheWAS analysis was conducted using data from the UK Biobank dataset (1283 diseases) and from FinnGen, showing a positive association with hypothyroidism risk and an inverse association with risk of goiter in both studies. An inverse association between the TSH PGS and thyroid cancer risk was also reported in the UK biobank, and replicated in the Columbus, deCODE and FinnGen studies.

At last, in order to evaluate the association between TSH levels and thyroid cancer risk using a two-sample mendelian randomization approach, an instrumental variable of TSH levels was derived from the 83 significant variants from the meta-analysis of HUNT and the ThyroidOmics consortium and its association with TC risk was tested in the meta-analysis of UKBB, MGI and deCODE. The association between the instrumental variable for TSH levels was also tested in association with goiter risk using the UKBB data. These analyses suggested a protective effect of high TSH levels on the risk of thyroid cancer and goiter.

This is a large GWAS study on TSH levels with analysis of its pleiotropic effects with thyroid conditions and disorders. The results are interesting, however I think some major information were missing making the article difficult to understand.

Here are some major comments:

1- Only data from HUNT and MGI are described. Description of deCODE, UK BB, FinnGen, Colombus and ThyroidOmics studies are missing. In particular, it would be important to describe how the phenotypes of interest (thyroid cancer, TSH levels, and thyroid disorders) were defined and how many individuals for each phenotype were available in these studies. If only summary statistics were available for these studies, this need to be mentioned.

We appreciate the reviewer's suggestion and added the description of the ThyroidOmics consortium, deCODE, UKBB, FinnGen and Columbus, USA in the Description of data sets subsection in the Methods section.

2- A descriptive table of the different studies with the number of cases for the phenotypes of interest and some indication on the age group and sex distribution would be helpful. Especially as age and sex have important effects on thyroid disorders.

We appreciate the reviewer's suggestion and we have added the information on age and sex distribution in the Description of data sets subsection in the Methods section.

3- If definition of thyroid conditions differs across the studies (self-declared vs validated cases, information on histology of thyroid cancer,...) this need to be commented further as limits of the study

in the discussion part. It is known that information on self-declared thyroid cancer can be very poor in some cohort studies, and also risk factors of medullary thyroid cancer are really different from papillary or vesicular thyroid cancer.

We would like to thank the reviewer for pointing this out. We have discussed this limitation in the discussion Section “Note that phenotype heterogeneity between the thyroid cancer datasets could have an impact on our results, although associations between higher TSH PGS and reduced thyroid cancer risk were observed in all four study populations. Different approaches were used to curate phenotypes for thyroid cancer, with cases being identified based on ICD codes mapped to phecodes in the UKBB^{5,6}, ICD codes combined with population registries in FinnGen and the Icelandic Cancer Registry in the Icelandic dataset⁷, while the papillary or follicular thyroid carcinoma cases were histologically confirmed in the Columbia, USA dataset⁷.”

4- In the TSH GWAS, data from HUNT and MGI studies were used. When several measurements of TSH levels were available for an individual, it is mentioned that in HUNT study, the last measurement was used, while in MGI study the authors used the average of the TSH levels. Why the authors did not use the same rule for both studies, this need to be justified/commented?

In the MGI study, TSH levels were measured as part of clinical care, and some TSH measurements may have been performed during acute illness that may have transiently increased or lowered TSH levels (the “non-thyroidal illness syndrome”). By using the average of the TSH measured, we aimed to minimize the influence of such transient changes. In contrast, the TSH measurements in HUNT were performed as part of a population survey, with the participants being less likely to be suffering from acute illness at the time of blood sampling. We apologize for the incorrect statement, when several measurements of TSH levels were available for an individual in HUNT, the first measurement was used, as measurements at an earlier age are likely to be less affected by unidentified health issues.

5- In general, the tables, supplementary tables and figures titles are not explicit and there not enough explanation in the legend. Abbreviations in tables should be clarified and meaning of some columns are not clear. For instance, I don't understand the column “direction” in Table 1.

We apologize for the confusion. We have added the explanation in legends or footnotes for columns. The column “direction” is the direction of effects of the testing alleles in each individual study.

6- The authors should explain the principle of the SuSiE method used for fine-mapping. Also, it is unclear how to interpret the results shown in Supplementary Table 4. A footnote with more explanation is required. Also, only 51 loci appeared in this table while 58 loci were highlighted by the GWAS. The authors should explain this difference.

We have added more details in the “Fine-mapping for potentially causal variants among TSH loci” subsection in the Results section. The reason why there are fewer loci than the total loci is that SuSiE failed to identify credible sets for causal signals for some loci as shown in Supplementary Table 4.

7- The authors should explain why they focused on TG genes and not on other loci while they conducted the fine-mapping analysis.

There are three main reasons why we focused on the gene TG. 1. The protein product of TG, thyroglobulin, is a key component of thyroid hormone. 2. TG is the most highly expressed gene in the

thyroid gland and its protein product represents roughly half the protein of the entire thyroid gland (PMID 26595189, our reference #41, PMID 30886364, our reference #40). 3. Two causal signals were identified in this locus and both contain a nonsynonymous variant that is nearly as significant as the top hits, making the functional experiments feasible to implement.

We have added a sentence to the second paragraph of the subsection “Fine-mapping for potentially causal variants among TSH loci ”

“TG encodes a highly specialized homodimeric multidomain glycoprotein for thyroid hormone biosynthesis²⁷ and it is the most highly expressed gene in the thyroid gland and its protein product represents roughly half the protein of the entire thyroid gland^{8,9}.”

8- In page 6, the authors explained why that they limited the analyses of pleiotropic effect on the 85 non HLA variants in the paragraph “Associations of PGS of TSH with other phenotypes”. This explanation should appear in the previous paragraph “Pleiotropic effects of TSH loci”. Also, it is not clear how many variants were removed from the analysis because they were located in the HLA region.

Thank you for pointing this out. We have added the explanation to the paragraph “Pleiotropic effects of TSH loci” (second sentence): “Due to the strong associations between HLA variants and autoimmune diseases³⁹, we excluded two HLA variants associated with TSH (rs1265091 and rs3104389) in the analysis for pleiotropic effects.” In the paragraph “Associations of polygenic scores of TSH with other phenotypes”: “As in the pleiotropy analysis, we excluded the two HLA variants in the PGS calculation to study the cumulative genetic effects of TSH-associated variants in non-HLA regions with human diseases.”

9- It seems that the authors are confused with the difference between case-control and cohort studies while citing the Columbus cohort as a “case-control cohort” (on page 7).

It does not make sense to look at the thyroid prevalence using the data from a case-control study. I do not understand the rationale to show both prevalence and OR of thyroid cancer in figure 2. If logistic regression were not adjusted on any factors, then both results (right and left panel) are similar but not on the same scale. Also, the results on prevalence are difficult to compare between studies. I suggest to remove the results on prevalence.

We agree that looking at the thyroid cancer prevalence using the data from a case-control study, such as the Columbus, USA study, may not be informative, so we have moved the plots for the Columbus, USA study to Supplementary Figure 10. As prevalence and OR have different scales and the magnitude of prevalence provides useful information for each data set, we decide to keep plots for prevalence and OR.

10- The authors should give the weight of the SNPs that permit to construct the instrumental variable for TSH levels, as it seems that it is derived from different studies (HUNT and the ThyroidOmics consortium) than for the polygenic score (HUNT study, MGI biobank and ThyroidOmics).

Also, they should explain why they did not use the same weight between the instrumental variable for TSH levels and the polygenic score. Also why FinnGen study was used to replicate the association between PGS and goiter but was not used for the MR analysis on goiter?

We apologize for the confusion and we have added statement to explain why weights from different GWASs for TSH were used in the paragraph “Mendelian Randomization for TSH, thyroid cancer and goiter”: “94 non-HLA genetic variants for TSH identified by our meta-analysis of HUNT, MGI and ThyroidOmics were used as instrumental variables (F-statistic for all SNPs > 23.72). To avoid sample overlap for GWASs of TSH and thyroid cancer, we used effects on TSH estimated by meta-analyzing HUNT and ThyroidOmics to construct the instrumental variable for TSH levels and for thyroid cancer, we meta-analyzed MGI, deCODE and UKBB. ” and “The effects on TSH were estimated by meta-analysis of HUNT, MGI and ThyroidOmics (Supplementary Table 1 and 2) and the GWAS results for goiter from UKBB were used.”

We have provided the weights of SNPs to construct instrumental variables for TSH levels in the MR analysis for TSH and thyroid cancer as Supplementary Table 16b and Supplementary Table 18c and 18d. FinnGen study was not used for the MR analysis on goiter because of the restricted data access to the summary statistics of goiter GWAS in FinnGen release 3.

11- 17 pleiotropic variants were reported between TSH levels and thyroid cancer risk. To my understanding, the instrumental variable included those variants in the mendelian randomisation (MR) analysis. As the presence of pleiotropic effect violates the condition of application of MR, a sensitivity analysis excluding these SNPs may be useful.

We detected variants that were associated with both TSH levels and thyroid cancer risk. However, this could be to vertical pleiotropy, i.e. that the variants influence thyroid cancer risk through influencing TSH levels, or horizontal pleiotropy, i.e. that the variants influence thyroid cancer risk through pathways other than TSH level. In the former case, selectively removing the TSH variants that are also statistically significantly associated with thyroid cancer risk would bias the MR results towards the null value. In the latter case, inclusion of pleiotropic variants could bias the MR results, and this is what we have investigated in several sensitivity analyses (with different assumptions regarding pleiotropic effects) including penalized weighted median, weighted median and weighted mode analyses (Figure 3a and Supplementary Table 16). Similar results were observed between methods, with the exception of the weighted mode which was strongly attenuated. Of note, the weighted mode has lower power than the inverse variance and weighted median methods, and it relies on a specific assumption termed ZEMPA (ZEro Modal Pleiotropy Assumption)¹⁰. For the information of the editor and reviewers, we have now additionally applied the MR Steiger test of directionality, to statistically test the assumption that the exposure causes the outcome. We found that the instrument explained more variance in the exposure than in the outcome, indicating that the correct causal direction is from TSH to thyroid cancer (p-value<0.001).

12- In the discussion part, the authors should discuss the results of the TSH GWAS in regards to the results from previous GWAS. Is there any loci that were previously highlighted that were not replicated in this new GWAS?

All previously reported loci for TSH have been replicated in this new meta-analysis with p-value < 5×10^{-8} . We have mentioned this in the Discussion section “Meta-analysis of the HUNT study, the MGI biobank and the ThyroidOmics consortium for TSH on up to 119,715 individuals identified 74 TSH loci, of which 28 are novel. All TSH loci that have been reported by previous GWAS studies have been successfully replicated in our meta-analysis¹¹⁻¹³.”.

13- In page 8, it is stated that the causal direction between TSH levels and thyroid cancer risk is unexpected. I think the author should mentioned and discuss recent findings from epidemiological

studies that showed an inverse association between TSH levels measures and papillary thyroid cancer. Here are some references:

-Franceschi S, Rinaldi S. TSH, Thyroid Hormone, and PTC-Letter. *Cancer Epidemiol Biomarkers Prev*. 2018 Feb;27(2):227.

- Huang H, Rusiecki J, Zhao N, Chen Y, Ma S, Yu H, et al. Thyroid-stimulating hormone, thyroid hormones, and risk of papillary thyroid cancer: a nested case-control study. *Cancer Epidemiol Biomarkers Prev* 2017;26:1209–18.

- Rinaldi S, Plummer M, Biessy C, Tsilidis KK, Ostergaard JN, Overvad K, et al. Thyroid-stimulating hormone, thyroglobulin, and thyroid hormones and risk of differentiated thyroid carcinoma: the EPIC study. *J Natl Cancer Inst* 2014.

We thank this reviewer for pointing this out and have incorporated these studies in the introduction.

Minor comments

1- Some studies were named differently along the manuscript. The Columbus study was names Ohio cohort in introduction, while deCODE study is sometime named Iceland study

2- I think that the description of the GWAS analysis should not be in the studies description but should appear separately to make it clearer.

3- The sentence “We performed genomic control correction was performed on HUNT and MGI...” “ (page 10) need to be corrected

4- Page 11: the “materials” section need to be rephrased

5- Order of table need to be revised, for instance Supp Table 13 is cited after Supp Table 14 in the text

We have made these changes.

Reviewer #3 (Remarks to the Author):

This study identifies 58 genome-wide significant loci for TSH levels, 17 of which are novel. These loci show pleiotropic effects on various diseases and traits, and provide genetic evidence for a causal relationship between TSH levels and the risk of thyroid cancer and goiter. Associations in the thyroglobulin (TG) locus are fine-mapped to two missense variants in TG, and functional work provides evidence for their role in thyroglobulin secretion.

This interesting study contains several novel findings that significantly expands the current genetic knowledge in this field. However, I have several comments/concerns presented below in the same order as presented in the manuscript (from Results to the Methods section):

1. Was there any evidence of inflation in the GWAS statistics due to reasons other than polygenicity (e.g. population structure)? It would be informative to investigate this using the univariate LD score regression intercept.

To account for sample relatedness, we conducted GWAS for inverse-normalized TSH in HUNT using a linear mixed model with random effects as implemented in SAIGE⁵. We have included genetic principal components (PCs) to account for population stratification. It is not appropriate to use the LD score regression intercept approach¹⁴ to evaluate the population stratification and polygenicity, as this method was not developed to use summary statistics from mixed models. In mixed models, the expected behavior of the expected value of Chi-square statistics is different from that in linear

regression model¹⁵. In addition, the linear assumption for the Chi-square statistics and LD scores by the LD score regression is not valid for TSH as shown in the plots below. The plot on the right is a scatter plot for Chi-square statistics from HUNT against LD scores for 1.3 million HapMap variants. The LD scores are pre-estimated by the program LDSC¹⁴ based on EUR samples in HapMap. The plot is overlaid with the mean of the Chi-square statistics by binned LD scores (yellow dots). The plot on the left is for variants with Chi-square statistics < 10 only. Both plots suggest that the genetic architecture of TSH does not fit the linear assumption by LDSC¹⁴.

2. Four rare variants (rs546738875, rs145153320, rs139352934, and rs121908872) that were associated with TSH levels in HUNT were not available in the ThyroidOmics dataset for replication. The authors explain that the frequency of the risk allele were somewhere between 11-22 times higher in HUNT than the other studies. However, how are the authors convinced that these results are not false positive findings in HUNT? In addition, were these variants genome-wide significant in the original analysis or only after performing the conditional analysis? If the latter, could this be an artifact due to the conditional analysis?

Thank you for pointing this out. These variants are genome-wide significant in the original analysis and have high imputation quality scores in HUNT (rs546738875: 0.94, rs145153320: 1, rs121908872: 0.98, rs139352934: 1). Direct genotyping would be needed to further confirm genotypes. We have mentioned this in the Discussion section “Additional independent signals were identified among several loci based on GWAS results in the meta-analysis and LD information in the HUNT study, including two rare variants rs546738875 and rs145153320 at the *B4GALNT3* locus and two rare missense variants *TSHR* p.A553T (rs121908872) and *TSHR* p.R609Q (rs139352934), which have been observed to be associated with congenital hypothyroidism in previous family studies^{2,3}. *TSHR* p.R609Q (rs139352934) is the most strongly associated with TSH in the thyroid stimulating hormone receptor gene *TSHR* with an effect size greater than one standard deviation of TSH (1.036 mU/L). As these rare variants were only imputed in HUNT, not in MGI or ThyroidOmics, further follow-up to verify the associations is needed.”

3. Were the significantly associated rare variants as well as those included in the SKAT-O analysis imputed? If so, was their imputation quality reasonably high? Please provide their imputation quality scores in a supplementary Table.

Yes, the significantly associated rare variants and those included in the SKAT-O analysis were imputed or directly genotyped. Only variants with imputation quality score ≥ 0.8 were included in the SKAT-O analysis. The imputation quality score for all significant top variants identified in single-variant association tests has been included in the Supplementary Tables 1 and 2.

4. The authors mention: "An intronic variant (rs6544660) in the thyroid adenoma associated gene THADA was recently identified to be associated with TSH". However, this variant has been reported as a novel discovery in this study in Table 1

We apologize for the confusion and we revised the sentence to be "An intronic variant (rs10186921) in the thyroid adenoma associated gene THADA was identified to be associated with TSH."

5. Were there any sample overlaps between the discovery and target datasets used for the PGS and MR analyses? I noticed that this study by design attempted to avoid such sample overlaps; however, it would be ideal to ensure that the results are not biased due to hidden sample overlaps or relatedness between individuals in the discovery and targets datasets. This can be investigated, for instance, using the bivariate LD score regression intercept.

We agree with the reviewer's concern and would like to evaluate the sample overlap if appropriate methods are available. However, the LD score regression approach was not developed for summary statistics from mixed models and as is shown in Q1, the linear assumption for LD score and Chi-square statistics is not valid for TSH GWAS in HUNT. Therefore, the bivariate LD score regression intercept will not be feasible to use here.

6. Apart from thyroid cancer and goiter, did the authors investigate causality of the other phenotypes that were associated with TSH PGS? Performing MR for the traits that were significantly associated in PGS analyses may help understand whether these genetic links are consistent with pleiotropy or causality. I do not have a strong preference whether all these analyses should be included in this paper; however, considering the PGS results, some reader may be interested to know whether the genetically predicted TSH levels are causally associated with any of the other traits that came up in the PGS analyses.

We agree with the reviewer that the PGS analyses hint to other traits and conditions that may be influenced by thyroid function, and although we chose to focus our MR analyses on thyroid cancer, MR analyses of all these outcomes would be interesting. However, such analyses and the collection of relevant outcome data (where available) for all these outcomes would greatly expand our study. We therefore feel that this is beyond the scope of the present study. Instead, our PGS analyses hint to possible causal effects that may be examined in depth in subsequent studies.

7. For the MR findings for thyroid cancer, although the CIs of the MR estimates overlap between the sensitivity tests, most of the test statistics are not significant (e.g. MR Egger, simple mode, and weighted mode in Suppl Table 15). Please discuss this in the text. Is this a power issue?

We agree with the reviewer's comment. Different sensitivity analysis for the MR results do have varying degrees of power, with MR-Egger being the most underpowered method and the weighted mode having lower power than the inverse variance and weighted median methods as it relies on a specific assumption termed ZEMPA (ZEro Modal Pleiotropy Assumption)¹⁰. Correspondingly, the P-values for these tests are unlikely to be significant with our sample size. However, we have drawn our conclusions based mainly on the direction and size of the effect, so not to be misled by this issue of power.

8. It would be helpful to discuss briefly, what would be the clinical implication of showing evidence of causality of higher TSH levels on reduced risk of thyroid cancer? Is there any evidence showing that increasing TSH levels is capable of treating cancer? Alternatively, in terms of prevention strategies, is increasing TSH levels practical, or can this lead to thyroid dysfunction and adverse metabolic consequences?

These concerns are relevant, and we have included them in the revised Discussion section: **“Although our results suggest that interventions elevating TSH levels could potentially reduce thyroid cancer risk, this suggestion lacks evidence from intervention studies. Further, any intervention that increases TSH levels by inhibiting thyroid hormone production would expectedly lead to symptoms and signs of hypothyroidism.”**

9. How well did the TSH PGS predict the phenotypes investigated? Providing r^2 or AUC in a supplementary table can be informative. Although I would expect that, similar to the other complex traits, a good prediction accuracy requires much larger sample sizes.

Thank you for this question. We have included r^2 for the TSH PGS for binary phenotypes in the UKBB in Supplementary Table 14 and added the following text in the paragraph for Associations of polygenic scores of TSH with other phenotypes: **“We also evaluated the phenotypic variance (r^2 on the liability scale) explained by TSH PGS for 596 phenotypes in the UKBB that have at least 500 cases in 280,943 unrelated white British samples (Supplementary Table 14). The phenotypes with highest r^2 were nontoxic nodular goiter ($r^2 = 0.96\%$), secondary hypothyroidism ($r^2 = 0.46\%$) and thyrotoxicosis with or without goiter ($r^2 = 0.16\%$).”**

10. The authors mention that the previously identified 47 independent TSH-associated loci account for 14% of TSH heritability. How much the novel loci add to the explained heritability? In addition, based on the other complex traits, explaining 14% of variance in heritability from only 47 SNPs seems to be too high. Is that because several common SNPs with large effect sizes are involved? Alternatively, please ensure that the explained heritability is estimated accurately.

Thanks for pointing this out. We now estimated the variance of TSH explained by TSH-associated loci as sum of effect size²*2*MAF*(1-MAF) for all independent top hits, where effect size is in the unit of SD of TSH. Using the same approach, previously identified 46 independent TSH-associated loci (merged two nearby variants that are located in one locus) accounted for 9.4% of TSH variance and together with the novel loci that have been identified by our study account for 13.3% variant of TSH. We have added a sentence at the end of the first paragraph of the RESULTS section **“In total, 99 independent top variants have been identified at the 74 loci, explaining 13.3% of the variance of TSH.”** We have added a subsection “Variance of TSH explained by loci” in the Methods section **“We estimated the variance of TSH explained by TSH-associated loci as sum of effect size² x 2 x MAF x (1-MAF) for all independent top hits, where effect size is in the unit of SD of TSH.”** Of note, the 9.4% of

the TSH variance explained by the 46 loci explains indeed 14% of the TSH heritability (which was estimated up to 65%). As the reviewer correctly assumed, this large number can be justified by several common SNPs with quite large effect sizes which were partly revealed already in former GWAS studies on TSH including relatively small sample sizes.

11. For the significant traits in PheWAS (e.g. for the 29 disease phenotypes), it would be interesting to know whether the genetic link with those traits also expand throughout the genome. The overall genetic correlation can be investigated using summary statistics using the LD score regression approach.

Thank you for the suggestion. As we have previously discussed in Q1 and Q5, as the linear assumption between Chi-square statistics and LD scores does not hold for TSH, it would not be appropriate to use LD score regression to estimate the genetic correlation between TSH and those disease phenotypes. In spite of this, we have estimated the genetic correlation with LDSC. As the plot below shows, hypothyroidism and goiter are diseases that have most significant p-values for genetic correlation with TSH and the directions are the same as their association with TSH PGS.

12. Although having thyroid gland as the most significant tissue in the DEPICT tissue enrichment analysis is expected and interesting, none of the tissue enrichment associations were statistically significant after correcting for multiple testing. I suggest specifying this in the text.

Thank you for the suggestion. Given that we have doubled our sample sizes for meta-analysis, as recommended by the DEPICT manual, we have re-run DEPICT using 162 loci with p-values < 1E-5 that are LD pruned and clumped based on TSH association p-values. Beside thyroid gland, we have seen more tissues in the tissue enrichment analysis and described the results in the subsection “Prioritization of TSH genes, pathways and tissues”. To further understand the biology underlying TSH associations, we prioritized associated genes as well as tissues and cell types in which TSH genes are likely to be highly expressed using DEPICT¹⁶ based on 161 loci with TSH association p-value cutoff 1×10^{-5} and clumped based on LD in HUNT. As expected, the membranes and thyroid gland are tissues most strongly associated followed by tissues from digestive system (ileum, gastrointestinal tract, pancreas and colon), respiratory system (lung) and accessory organs for eyes (conjunctiva, eyelids, and anterior eye) (Supplementary Table 6). Based on functional similarity to other genes among TSH loci, 70 genes at the TSH associated loci were prioritized by DEPICT with false discovery rate (FDR) ≤ 0.01 (Supplementary Table 7), among which the prioritized genes *ZFP36L2*, *B4GALNT3*, *PPP1R3B*,

***FAM109A, GNG12, GADD45A, BMP2, VEGFC, LPP and MAL2* were at the novel TSH loci identified in our meta-analysis (Table 1). Additionally, among 14,461 reconstituted gene sets, 56 gene sets were enriched among TSH loci with $FDR < 0.01$. The most significantly enriched one is CTSD PPI subnetwork, followed by genes sets for regulation of phosphorylation (Supplementary Table 8)."**

13. To get an unbiased understanding of genetic link between TSH and thyroid cancer, instead of using only SNPs that were also significant for thyroid cancer, it was better to include all the significant TSH SNPs in the analysis presented in supplementary Fig 6 as well as binomial tests for the direction of effect. In addition, adding correlation coefficient or variance explained to the supplementary Fig 6 would be helpful. Having said that, the caveat for using all the TSH significant SNPs in the binomial tests for the direction of effect would be that some SNPs might show unexpected direction of effect for thyroid cancer while their CIs overlap with the expected direction.

We have included a plot with all significant TSH variants in the Supplementary Figure 6 and described the results at the end of the subsection Pleiotropic effects of TSH loci: "We further examined the association with thyroid cancer for TSH index variants. We meta-analyzed deCODE⁷ and UKBB^{4,5} for thyroid cancer in 3,359 thyroid cancer cases and 694,949 controls and examined 94 out of 99 TSH non-HLA index variants that are available in the meta-analysis for thyroid cancer (Supplementary Table 11). The TSH increasing alleles of 63 out of 94 TSH-associated variants (67%) were associated with reduced thyroid cancer risk (Supplementary Table 11 and Supplementary Figure 6a and 6b) ($P\text{-value}_{\text{binomial}} = 1.26 \times 10^{-3}$). 18 out of the 94 TSH-associated variants tested (19%) were at least nominally associated with thyroid cancer ($P < 0.05$) ($P\text{-value}_{\text{binomial}} = 1.18 \times 10^{-9}$). For 16 out of the 18 TSH-associated variants, the TSH increasing alleles were associated with reduced thyroid cancer risk ($P\text{-value}_{\text{binomial}} = 1.31 \times 10^{-3}$, Supplementary Table 11 and Supplementary Figure 6c and 6d). Moreover, when we examined alleles that predisposed to thyroid cancer^{7,12,17}, 9 out of 11 had a consistent direction of effect towards lower TSH ($P\text{-value}_{\text{binomial}} = 0.065$). Of the six thyroid cancer risk-alleles that were at least nominally associated with TSH level ($P < 0.05$), all six variants were associated with lower TSH ($P\text{-value}_{\text{binomial}} = 0.03$) (Supplementary Table 12 and Supplementary Figure 7)."

14. The authors have inversed normalized TSH levels for each study separately, and report the association results on the SD scale per study. Was the SD similar across all the studies? Different SD across studies makes it difficult to convert the meta-analyzed effect estimates back on the actual TSH levels scale.

Thank you for the pointing this out. The mean of TSH in HUNT is 1.659 mU/L and SD is 1.036 mU/L, the mean and SD for TSH in MGI are 1.914 mU/L and 1.175 mU/L. The ThyroidOmics consortium contains 22 study and the reference range of TSH levels for each study can be found in Table S1 in Teumer *et al.* 2018¹³. We agree with the reviewer that one drawback of the meta-analysis for quantitative traits that are inverse-normal transformed within individual data set is that it is difficult to convert the effect estimates in the meta-analysis back to original scale of TSH levels. We have mentioned this limitation at the end of the second paragraph of the DISCUSSION section "As individual GWAS was conducted on inverse-normal transformed TSH levels before meta-analysis, it is challenging to convert the effect sizes reported by our meta-analysis to actual scales of TSH levels."

15. The investigators specify that they have used different methods for measuring TSH levels across studies. Does this bias the results? Do they get similar results after adding the measurement method as a covariate in their regression analysis?

In the updated GWAS for the HUNT study, we have controlled for laboratory method although the two measurement methods for TSH in HUNT2 and HUNT3 have been previously shown to yield similar measurements (PMID: 23975540, our reference #54). We have used a covariate “TSH batch” to indicate whether the TSH was measured in HUNT2 or HUNT3 in the GWAS and included the description of the GWAS model that has been used in HUNT in the Description of data sets subsection in the METHODS section “We used TSH batch (indicating whether TSH was measured in HUNT2 or HUNT3), genotype batch, the first fifteen genetic principal components (PCs), which account for 85% of the genetic variance among study samples, age at the measurement of TSH and sex as covariates.”.

In all three data sets: HUNT, MGI and ThyroidOmics, GWAS were conducted on inverse-normalized TSH levels. We have also conducted the Cochran's Q-test for heterogeneity of TSH loci that were identified in our meta-analysis as shown in Table 1 and Supplementary Table 1 and 2. All novel loci for TSH have the heterogeneity P-values greater than 0.05, indicating that heterogeneity of genetic effects of those loci was not detected across the three data sets.

16. The authors mention that “We computed polygenic score (PGS) for TSH based on the 84 non-HLA significant index variants weighted by the effect size estimates based on the meta-analysis results of the HUNT Study and the MGI Biobank and then with the GWAS results from the ThyroidOmics consortium”. Does this mean that PGS was calculated from the HUNT + MGI + ThyroidOmics meta-analysis, or was PGS based on ThyroidOmics estimates calculated separately for a comparison?

We apologize for the unclear description. As we have updated our analysis by meta-analyzing HUNT (N=55,342), MGI (N = 10,085), and ThyroidOmics (N up to 54,288) together, the PGS for TSH was constructed based on the 95 independent variants identified in the meta-analysis of the three data sets. We have described this in the Associations of polygenic scores of TSH with other phenotypes subsection of the RESULTS section “While individual TSH variants may exhibit pleiotropic effects, it is also possible that the cumulative effects of TSH-modifying genetic variants may lead to disease. Therefore, we constructed polygenic scores (PGS) from the 95 independent non-HLA TSH top variants (rs1265091 and rs3104389 are HLA variants and rs121908872 and 23:3612081 were not in UKBB) and examined their association with the 1,283 human diseases constructed from ICD codes in the UKBB^{4,5,18}.”

17. For HUNT, the authors have noted that if multiple TSH measurements were available, they used the most recent one. However, for MGI they used the average of the TSH levels in the same scenario. Was this inconsistency justified, or is it due to different analysis plans per study prior to conducting the meta-analysis?

In the MGI study, TSH levels were measured as part of clinical care, and some TSH measurements may have been performed during acute illness that may have transiently increased or lowered TSH levels (the “non-thyroidal illness syndrome”). By using the average of the TSH measured, we aimed to minimize the influence of such transient changes. In contrast, the TSH measurements in HUNT were performed as part of a population survey, with the participants being less likely to be suffering from acute illness at the time of blood sampling. We apologize for the incorrect statement, when several measurements of TSH levels were available for an individual, the first measurement was used, as measurements at an earlier age may be less affected by unidentified health issues.

18. It was not clear why the authors have only included significant variants from the meta-analysis of

HUNT + MGI in their DEPICT analysis. Would the final meta-analyzed results including ThyroidOmics dataset be more powerful for their DEPICT analysis?

In our current analysis for DEPICT, the loci were defined by LD pruning based on LD information in the HUNT study and clumping by P-values from the meta-analysis of HUNT, MGI, and ThyroidOmics.

19. Please provide details of the stepwise conditional analysis in the Methods section. Was this done using GWAS summary statistics or raw genotypes?

The stepwise conditional analysis was done using GWAS summary statistics. We have provided more details of the analysis approach in the Stepwise conditional analysis subsection of the Methods section “We identified additional independent association signals at each locus using a stepwise conditioning approach in GCTA-COJO¹⁹(--cojo-slc) and reported top variants with conditional P-values $\leq 5 \times 10^{-8}$.”

20. The authors say: “For all genetic variants with P-value < 5×10^{-6} , we conducted another meta-analysis with up to 54,288 additional samples in ThyroidOmics”. Why the meta-analysis was not performed using all the SNPs in these three studies? Was this because of access restrictions to ThyroidOmics data?

In our initial meta-analysis approach, we included the results from the ThyroidOmics consortium in the replication step only. In the revised manuscript, we implemented the more powerful joint GWAS meta-analysis of all three datasets for locus discovery which was also suggested by Reviewer #1. Thus, we have updated our analysis by meta-analyzing HUNT (N=55,342), MGI (N = 10,085), and the ThyroidOmics consortium (up to N = 54,288) together.

References

- 1 Genomes Project, C. *et al.* A global reference for human genetic variation. *Nature* **526**, 68-74, doi:10.1038/nature15393 (2015).
- 2 Calebiro, D. *et al.* Frequent TSH receptor genetic alterations with variable signaling impairment in a large series of children with nonautoimmune isolated hyperthyrotropinemia. *J Clin Endocrinol Metab* **97**, E156-160, doi:10.1210/jc.2011-1938 (2012).
- 3 Abramowicz, M. J., Duprez, L., Parma, J., Vassart, G. & Heinrichs, C. Familial congenital hypothyroidism due to inactivating mutation of the thyrotropin receptor causing profound hypoplasia of the thyroid gland. *J Clin Invest* **99**, 3018-3024, doi:10.1172/JCI119497 (1997).
- 4 Bycroft, C. *et al.* The UK Biobank resource with deep phenotyping and genomic data. *Nature* **562**, 203-209, doi:10.1038/s41586-018-0579-z (2018).
- 5 Zhou, W. *et al.* Efficiently controlling for case-control imbalance and sample relatedness in large-scale genetic association studies. *Nat Genet* **50**, 1335-1341, doi:10.1038/s41588-018-0184-y (2018).
- 6 Denny, J. C. *et al.* Systematic comparison of phenome-wide association study of electronic medical record data and genome-wide association study data. *Nat Biotechnol* **31**, 1102-1110, doi:10.1038/nbt.2749 (2013).

- 7 Gudmundsson, J. *et al.* A genome-wide association study yields five novel thyroid cancer risk loci. *Nat Commun* **8**, 14517, doi:10.1038/ncomms14517 (2017).
- 8 Citterio, C. E., Targovnik, H. M. & Arvan, P. The role of thyroglobulin in thyroid hormonogenesis. *Nat Rev Endocrinol* **15**, 323-338, doi:10.1038/s41574-019-0184-8 (2019).
- 9 Di Jeso, B. & Arvan, P. Thyroglobulin From Molecular and Cellular Biology to Clinical Endocrinology. *Endocr Rev* **37**, 2-36, doi:10.1210/er.2015-1090 (2016).
- 10 Hartwig, F. P., Davey Smith, G. & Bowden, J. Robust inference in summary data Mendelian randomization via the zero modal pleiotropy assumption. *Int J Epidemiol* **46**, 1985-1998, doi:10.1093/ije/dyx102 (2017).
- 11 Malinowski, J. R. *et al.* Genetic variants associated with serum thyroid stimulating hormone (TSH) levels in European Americans and African Americans from the eMERGE Network. *PLoS One*. **9**, e111301 (2014).
- 12 Gudmundsson, J. *et al.* Discovery of common variants associated with low TSH levels and thyroid cancer risk. *Nat Genet* **44**, 319-322, doi:10.1038/ng.1046 (2012).
- 13 Teumer, A. *et al.* Genome-wide analyses identify a role for SLC17A4 and AADAT in thyroid hormone regulation. *Nat Commun* **9**, 4455, doi:10.1038/s41467-018-06356-1 (2018).
- 14 Bulik-Sullivan, B. K. *et al.* LD Score regression distinguishes confounding from polygenicity in genome-wide association studies. *Nat Genet* **47**, 291-295, doi:10.1038/ng.3211 (2015).
- 15 Yang, J., Zaitlen, N. A., Goddard, M. E., Visscher, P. M. & Price, A. L. Advantages and pitfalls in the application of mixed-model association methods. *Nat Genet* **46**, 100-106, doi:10.1038/ng.2876 (2014).
- 16 Pers, T. H. *et al.* Biological interpretation of genome-wide association studies using predicted gene functions. *Nat Commun* **6**, 5890, doi:10.1038/ncomms6890 (2015).
- 17 Gudmundsson, J. *et al.* Common variants on 9q22.33 and 14q13.3 predispose to thyroid cancer in European populations. *Nat Genet* **41**, 460-464, doi:10.1038/ng.339 (2009).
- 18 Wu, P. *et al.* Developing and Evaluating Mappings of ICD-10 and ICD-10-CM Codes to PheCodes. *JMIR Med Inform.* **7(4):e14325**, doi:10.2196/14325. (2019).
- 19 Yang, J. *et al.* Conditional and joint multiple-SNP analysis of GWAS summary statistics identifies additional variants influencing complex traits. *Nat Genet* **44**, 369-375, S361-363, doi:10.1038/ng.2213 (2012).

REVIEWERS' COMMENTS:

Reviewer #1 (Remarks to the Author):

I would like to thank the Authors for fully addressing all issues and explaining the points that were unclear to me.

Reviewer #2 (Remarks to the Author):

The authors have satisfactorily responded to the questions of the reviewers and made the necessary changes to the manuscript.

Minor comment:

There is a discordance between the number of cases and controls indicated in the text ("3,359 thyroid cancer cases and 694,949 controls") and in the Supp Table 11 ("1,361 cases and 686,390 controls") for thyroid cancer analysis using DeCode and UKB. Also p-values for heterogeneity and direction of effects are missing in the table.

Reviewer #3 (Remarks to the Author):

The authors have adequately addressed my concerns. I have only couple of minor points below:

I agree with the authors that the LDSC regression approach was not developed for mixed models. However, several studies have successfully used this approach on the GWAS summary statistics obtained from linear mixed models to investigate inflation of the test statistics due to relatedness or population structure. For example, please see Lou et al, Nat Genet 2018 (<https://www.ncbi.nlm.nih.gov/pmc/articles/PMC6309610/>) which uses attenuation ratio defined as $(LDSC\ intercept - 1) / (\text{mean } \chi^2 - 1)$ for the summary statistics obtained from linear mixed model in Bolt LMM. However, because the author have shown that the linearity assumption in the LDSC regression approach does not hold for their TSH GWAS, I agree that it will not be appropriate to use LDSC regression for their study.

Thanks for including r^2 estimates for the PGS analyses. I noticed that the authors have specified that "We also evaluated the phenotypic variance (r^2 on the liability scale) explained by TSH PGS...". As age, sex, and PCs were also included as covariates in the PGS analyses, the r^2 should refer to the variance explained by all these variables collectively? Or was r^2 estimated for the PGS variable alone? If the former, I suggest re-phrasing the sentence to show that the r^2 estimates are for the full model.

For the DEPICT tissue enrichment analysis, still none of the tissues show significant results after correction for multiple testing ($P < 0.05/208$ tests) or $FDR < 0.05$. I think when the authors talk about the top tissues in the manuscript (e.g. tissues from the digestive system, etc.), it worth reminding the readers that none of these tissues passes the significance threshold.

Response to Reviewers Comments

Title: GWAS of thyroid stimulating hormone highlights its pleiotropic effects and genetic association with low thyroid cancer risk

We thank the reviewers and editors again for their thoughtful and constructive comments that helped us improve the manuscript. Below are our detailed responses (in bold).

Reviewer #1 (Remarks to the Author):

I would like to thank the Authors for fully addressing all issues and explaining the points that were unclear to me.

We thank the reviewer for the positive evaluation of our work.

Reviewer #2 (Remarks to the Author):

The authors have satisfactorily responded to the questions of the reviewers and made the necessary changes to the manuscript.

Minor comment:

There is a discordance between the number of cases and controls indicated in the text ("3,359 thyroid cancer cases and 694,949 controls") and in the Supp Table 11 ("1,361 cases and 686,390 controls") for thyroid cancer analysis using DeCode and UKB. Also p-values for heterogeneity and direction of effects are missing in the table.

We appreciate the reviewer's positive assessment of our work. We apologize for the inconsistent sample sizes for thyroid cancer. In the Supplementary Table 11, the sample sizes have been updated to "3,359 thyroid cancer cases and 694,949 controls". Note that as the meta-analysis was done using UKBB and a meta-analysis for thyroid cancer (3,001 cases and 287,550 controls) with deCODE and four other case-control data sets with European ancestry as previously reported by Gudmundsson et al. (2017) (reference #19 in the manuscript), we replace deCODE with Gudmundsson et al. (2017) throughout the manuscript to avoid confusion.

We have added the P-values for heterogeneity and direction of effects in the Supplementary Data 9 (previous Supplementary Table 11).

Reviewer #3 (Remarks to the Author):

The authors have adequately addressed my concerns. I have only couple of minor points below:

I agree with the authors that the LDSC regression approach was not developed for mixed models. However, several studies have successfully used this approach on the GWAS summary statistics

obtained from linear mixed models to investigate inflation of the test statistics due to relatedness or population structure. For example, please see Lou et al, Nat Genet 2018 (<https://www.ncbi.nlm.nih.gov/pmc/articles/PMC6309610/>) which uses attenuation ratio defined as $(LDSC\ intercept - 1) / (\text{mean } \chi^2 - 1)$ for the summary statistics obtained from linear mixed model in Bolt LMM. However, because the author have shown that the linearity assumption in the LDSC regression approach does not hold for their TSH GWAS, I agree that it will not be appropriate to use LDSC regression for their study.

We thank for the reviewer's comments. We agree that the attenuation ratio can be used for the summary statistics obtained from linear mixed model as long as the linearity assumption holds for the data set.

Thanks for including r^2 estimates for the PGS analyses. I noticed that the authors have specified that "We also evaluated the phenotypic variance (r^2 on the liability scale) explained by TSH PGS...". As age, sex, and PCs were also included as covariates in the PGS analyses, the r^2 should refer to the variance explained by all these variables collectively? Or was r^2 estimated for the PGS variable alone? If the former, I suggest re-phrasing the sentence to show that the r^2 estimates are for the full model.

We apologize for the confusion. r^2 was estimated for the PGS variable alone. More specifically, the Nagelkerke's r^2 was estimated. We have revised this sentence to be "We also evaluated the phenotypic variance (Nagelkerke's r^2)¹ explained by TSH PGS for 596 phenotypes in the UKBB that have at least 500 cases in 280,943 unrelated white British samples (Supplementary Data 12)." We have added the one sentence at the end of the "Polygenic score" subsection in the METHODS section to describe how the Nagelkerke's r^2 was estimated "The Nagelkerke's r^2 of TSH PGS for phenotypes in the UKBB was estimated as the difference of r^2 in the full models with PRS and non-genetic covariates birth year, sex and first 4 genetic PCs and r^2 in the reduced models with non-genetic covariates only without PRS using the R library rcompanion."

For the DEPICT tissue enrichment analysis, still none of the tissues show significant results after correction for multiple testing ($P < 0.05/208$ tests) or $FDR < 0.05$. I think when the authors talk about the top tissues in the manuscript (e.g. tissues from the digestive system, etc.), it worth reminding the readers that none of these tissues passes the significance threshold.

We thank the reviewer for pointing this out. We have added this piece of information to the sentence in the paragraph for DEPICT results "As expected, the membranes and thyroid gland are the most strongly associated tissues followed by tissues from the digestive system (ileum, gastrointestinal tract, pancreas and colon), respiratory system (lung) and accessory organs for eyes (conjunctiva, eyelids and anterior eye), although none of the tissues reached the Bonferroni significant threshold (P -value $< 0.05/209$ tests) or have false discovery rate (FDR) < 0.05 "

1 Nagelkerke, N. J. D. A Note on a General Definition of the Coefficient of Determination. *Biometrika* **78**, 691-692, doi:10.2307/2337038 (1991).